# Noisy-Space Policy Gradient for Diffusion Policies in Offline Reinforcement Learning

**Mahmoud Selim** [1,2]  **Cristina Cipriani** [1]  **Karl H. Johansson** [2]

## Abstract

Diffusion policies offer a powerful and expressive parameterization for continuous control. Yet, their integration with reinforcement learning remains conceptually and algorithmically challenging. In this work, we address this gap by introducing a noisy-space action-value (Q-)function that assigns values to diffusion latents through the distribution of executed actions induced by the denoising process. We show that this construction admits a precise semantic interpretation and derive a noisy-space policy gradient (NSPG) that optimizes noisy latents using only clean action-space value estimates. Building on this result, we formulate a KL-regularized policy improvement over noisy latents and show that the resulting objective admits a diffusion-compatible regression form, avoiding backpropagation through the denoising process. Empirical results on state-based D4RL benchmarks and vision-based OGBench tasks demonstrate that the proposed noisy-space objective provides a principled and effective basis for training diffusion policies in offline reinforcement learning. Project webpage: `https://mahmoud-selim.github.io/NSPG/`

## 1. Introduction

Offline reinforcement learning (RL) aims to learn effective decision-making policies from previously collected datasets, without requiring additional interaction with the environment (Lange et al., 2012; Levine et al., 2020). In this setting, policy optimization is inherently constrained: the learned policy must improve performance while remaining within the support of the data distribution, as extrapolating beyond observed state–action pairs can lead to severely biased value estimates (Kumar et al., 2019). In practice, offline datasets are often heterogeneous and exhibit complex, multimodal action distributions arising from mixtures of behaviors, suboptimal demonstrations, or task variability (Collaboration & Abby O'Neill, 2023). This has motivated growing interest in expressive generative policy classes capable of representing such distributions in offline reinforcement learning.

Among expressive generative policy classes, diffusion-based policies (Janner et al., 2022; Chen et al., 2024; Wang et al., 2022) have recently gained traction due to their ability to represent rich, multimodal action distributions through iterative denoising dynamics. Unlike conventional parametric policies that map states directly to actions (Hansen-Estruch et al., 2023; Kumar et al., 2020), diffusion policies generate actions by optimizing and sampling through a sequence of noisy latent variables, gradually transforming an initial noise sample into a final action via a reverse-time diffusion process (Ho et al., 2020; Sohl-Dickstein et al., 2015).

This design decouples action generation from a single feedforward decision, instead distributing it across multiple noise levels that are not directly observable by the environment. As a consequence, policy optimization and sampling take place in a noisy latent space, while reinforcement learning objectives—such as value functions, advantages, and policy gradients—are defined over the executed, noise-free actions that interact with the environment. This structural separation introduces an intrinsic mismatch between the space in which optimization is performed and the space in which value is defined, raising the question of how to define and optimize value-based objectives for diffusion policies in a principled and semantically coherent manner.

Existing diffusion-based RL methods address the latent–action mismatch through distinct strategies, but none provide a principled resolution. Some treat diffusion models purely as action samplers (Chen et al., 2022), selecting or reweighting generated actions using an action-space critic, thereby bypassing latent-space guidance altogether. Others optimize diffusion policies end-to-end by backpropagating through the denoising process (Wang et al., 2022), coupling policy improvement to long stochastic computation graphs with substantial computational overhead and instability. Recent variants improve efficiency by simplifying

[1]TRATON CV AB, Stockholm, Sweden [2]KTH Royal Institute of Technology, Stockholm, Sweden. Correspondence to: Mahmoud Selim <mahmoud.selim@scania.com>.

*Proceedings of the 43rd International Conference on Machine Learning*, Seoul, South Korea. PMLR 306, 2026. Copyright 2026 by the author(s).

the generative model (e.g., one-step or distilled predictors), trading expressiveness for tractability (Park et al., 2025). Finally, methods that inject value guidance directly in noisy latent space evaluate Q-functions on intermediate noisy variables (Fang et al., 2024), even though these variables are not the executed actions that interact with the MDP. Across all approaches, the relationship between noisy latents and action-space value is either avoided or imposed using the clean action value function, rather than defined through an explicit model.

This gap calls for a value formulation that bridges noisy latent-space optimization and action-space returns. We therefore define a noisy-space action-value function $Q_{noisy}$ as the expected action-space return induced by the diffusion policy conditioned on a given noisy latent variable. This construction assigns value to noisy latents through the distribution of clean actions they generate under the denoising process, ensuring that the critic is evaluated only on clean actions rather than on intermediate noisy variables.

Building on this formulation, we derive a noisy-space policy gradient (NSPG) that directly optimizes the noisy-space action-value function. The resulting gradient yields a principled latent-space policy-improvement direction while avoiding backpropagation through the full denoising trajectory. We instantiate this objective in Noisy-Space Actor–Critic (NSAC), a practical actor–critic algorithm for stable value-based training of expressive diffusion policies. The contributions of this work can be summarized as follows:

- We identify a fundamental gap in value-based optimization for diffusion policies in RL, arising from the mismatch between noisy latent-space policy representations and action-space value functions.

- We introduce a noisy-space action-value function $Q_{noisy}$ that maps diffusion latent states to expected action-space returns. Moreover, we derive a noisy-space policy gradient (NSPG) for diffusion policies based on this formulation, enabling value-based learning without evaluating critics on noisy variables or backpropagating through the denoising process.

- We develop Noisy-Space Actor–Critic (NSAC), a stable and principled actor–critic algorithm for training diffusion policies directly in noisy latent space.

- We evaluate NSAC across state-based D4RL benchmarks and vision-based OGBench tasks, demonstrating consistent performance gains over prior methods.

## 2. Preliminaries

**Reinforcement Learning.** We consider an infinite-horizon discounted Markov Decision Process (MDP) $\mathcal{M} =$

$(\mathcal{S}, \mathcal{A}, P, r, \rho_0, \gamma)$ (Sutton & Barto, 2018). The state space is $\mathcal{S} \subseteq \mathbb{R}^{d_s}$ and the action space is $\mathcal{A} \subseteq \mathbb{R}^{d_a}$, with dimensions $d_s$ and $d_a$, respectively. The transition kernel is $P(\cdot \mid s, a)$ over $\mathcal{S}$, the reward function is $r : \mathcal{S} \times \mathcal{A} \to \mathbb{R}$, $\rho_0$ denotes the initial-state distribution, and $\gamma \in (0, 1)$ is the discount factor. A stochastic policy $\pi(a \mid s)$ induces trajectories via $s_0 \sim \rho_0$, $a_t \sim \pi(\cdot \mid s_t)$, and $s_{t+1} \sim P(\cdot \mid s_t, a_t)$.

For a policy $\pi$, the action-value function $Q^\pi : \mathcal{S} \times \mathcal{A} \to \mathbb{R}$ is $Q^\pi(s, a) \triangleq \mathbb{E}_\pi[\sum_{t=0}^\infty \gamma^t r(s_t, a_t) \mid s_0 = s, \ a_0 = a]$, which represents the expected discounted return obtained by executing action $a$ in state $s$ and then following $\pi$ thereafter. Crucially, the action-value function is defined over *executed actions* $a \in \mathcal{A}$, which are the only variables that affect transitions and rewards in the MDP.

**Offline Reinforcement Learning.** Offline RL learns a policy from a fixed dataset $\mathcal{D} = \{(s_i, a_i, r_i, s_i')\}_{i=1}^N$ collected by an unknown behavior policy, without additional interaction with the environment (Levine et al., 2020). Because value estimation is restricted to the support of $\mathcal{D}$, policy improvement is typically formulated as a constrained optimization problem (Nair et al., 2020):

$$\pi_{k+1} = \arg\max_\pi \ \mathbb{E}_{s \sim \mathcal{D}}\Big[\mathbb{E}_{a \sim \pi(\cdot|s)} Q^{\pi_k}(s, a)\Big]$$

$$\text{s.t. } \mathbb{E}_{s \sim \mathcal{D}}\Big[\text{KL}\big(\pi(\cdot|s) \,\|\, \pi_\beta(\cdot|s)\big)\Big] \le \epsilon_b, \quad (1)$$

where $\pi_\beta$ denotes the (unknown) behaviour policy.

**Diffusion Policies.** We represent policies using conditional diffusion models that generate actions via an iterative denoising process (Sohl-Dickstein et al., 2015; Ho et al., 2020; Song et al., 2022). Let $T \in \mathbb{N}$ denote the number of diffusion steps and $\{x^{(k)}\}_{k=0}^T \subset \mathbb{R}^{d_a}$ a sequence of latent variables with the same dimension as the action space. Action generation begins by sampling $x^{(T)} \sim p_T(\cdot)$, where $p_T$ is a fixed prior (e.g., a standard Gaussian), and applying a reverse-time denoising process conditioned on the state $s$:

$$x^{(k-1)} \sim p_\theta(x^{(k-1)} \mid x^{(k)}, s, k), \qquad k = T, \dots, 1. \quad (2)$$

where $\theta$ denotes the parameters of the diffusion policy. This yields a stochastic trajectory $x^{(T)} \to \cdots \to x^{(0)}$, where the final latent is mapped to the executed action $a = x^{(0)} \in \mathcal{A}$ (or via a deterministic map $a = g(x^{(0)}, s)$). The reverse process is trained to invert a forward noising process $q(x^{(k)} \mid a)$ that progressively corrupts clean actions. Given a noise schedule $\{\beta_k\}_{k=1}^T$, the forward process is

$$q(x^{(k)} \mid x^{(k-1)}) = \mathcal{N}\Big(x^{(k)}; \sqrt{1 - \beta_k}\, x^{(k-1)}, \beta_k I\Big), \quad (3)$$

which induces the marginal

$$q_k(x^{(k)} \mid a) = \mathcal{N}\Big(x^{(k)}; \sqrt{\bar{\alpha}_k}\, a, \, (1 - \bar{\alpha}_k)I\Big), \quad (4)$$

where $\alpha_k = 1 - \beta_k$ and $\bar{\alpha}_k = \prod_{i=1}^k \alpha_i$. For $k > 0$, the variables $x^{(k)}$ are referred to as *noisy latent variables*: although they lie in $\mathbb{R}^{d_a}$, they are not elements of the MDP action space $\mathcal{A}$ and are never directly applied to the environment. Only the final action $a$ interacts with the environment and determines rewards and transitions.

## 3. Noisy-Space Value Functions and Policy Optimization

In this section, we develop a principled framework for value-based optimization of diffusion policies in offline RL. We first define a noisy-space action-value function that serves as the objective for latent-space policy optimization. We then introduce a surrogate semantic model that renders this objective well-defined without assigning intrinsic MDP semantics to latent variables. Using this formulation, we derive a noisy-space policy gradient that operates directly in noisy latent space. Finally, we describe NSAC, an efficient algorithm that implements this gradient without backpropagation through the denoising process.

### 3.1. Noisy-Space Action-Value Function

As established in the preliminaries, diffusion policies generate actions through a sequence of noisy latent variables $\{x^{(k)}\}_{k=0}^T$, while value functions in reinforcement learning are defined with respect to executed actions $a \in \mathcal{A}$. To enable value-based optimization in latent space, we define a value function that associates each noisy latent state with the expected action-space return induced by the policy.

**Definition 3.1** (Noisy-Space Action-Value Function). Let $\pi_\theta$ be a diffusion policy and let $Q^{\pi_\theta}(s,a) : \mathcal{S} \times \mathcal{A} \to \mathbb{R}$ denote its action-value function. For a state $s \in \mathcal{S}$, a diffusion timestep $k \in \{0, \dots, T\}$, and a noisy latent variable $x^{(k)} \in \mathbb{R}^{d_a}$, we define the noisy-space action-value function

$$Q^{\pi_\theta}_{\text{noisy}}(s, x^{(k)}, k) \triangleq \mathbb{E}_{a \sim \pi_\theta(\cdot | s, x^{(k)}, k)}[Q^{\pi_\theta}(s,a)], \quad (5)$$

where the expectation is taken over the distribution of executed actions induced by the diffusion policy when conditioned on $(s, x^{(k)}, k)$.

This definition assigns value to a noisy latent state only through the actions it generates, and not through the latent variable itself. In particular, $Q^{\pi_\theta}_{\text{noisy}}$ is not a value function on noisy variables in the MDP sense, but a conditional expectation of the action-space value under the diffusion policy. As a result, $Q^{\pi_\theta}_{\text{noisy}}$ provides a well-defined objective for policy optimization in noisy latent space.

We next establish that $Q_{noisy}$ introduced in Definition 3.1 admits a precise semantic interpretation that is fully consistent with the environment interface of a diffusion policy.

**Proposition 3.2** (Semantic Correctness). *Let $\pi_\theta$ be a diffusion policy and let $Q^{\pi_\theta}_{\text{noisy}}$ be defined as in Definition 3.1.*

*For any state $s \in \mathcal{S}$, diffusion step $k \in \{0, \dots, T\}$, and noisy latent variable $x^{(k)} \in \mathbb{R}^{d_a}$, we have*

$$Q^{\pi_\theta}_{\text{noisy}}(s, x^{(k)}, k) = \mathbb{E}\left[\sum_{t=0}^\infty \gamma^t r(s_t, a_t) \;\middle|\; s_0 = s, \; x^{(k)}\right],$$
$$(6)$$

*where the initial action $a_0$ is sampled from $\pi_\theta(\cdot \mid s_0, x^{(k)}, k)$ and subsequent actions $a_t$ are sampled from the marginal policy induced by $\pi_\theta$.*

*Proof.* The proof is provided in the Appendix. □

Proposition 3.2 establishes that $Q^{\pi_\theta}_{\text{noisy}}(s, x^{(k)}, k)$ exactly equals the expected discounted return obtained when the diffusion policy is initialized at $(s, x^{(k)}, k)$ and subsequently executed in the environment. Since the environment interacts with the policy exclusively through executed actions, this characterization uniquely determines the value of any function defined on noisy latent variables that is consistent with the environment semantics. In particular, any value-based objective for diffusion policies that depends on $(s, x^{(k)}, k)$ and agrees with the environment return must coincide with $Q^{\pi_\theta}_{\text{noisy}}$.

### 3.2. Surrogate Semantics for Noisy Latents

Definition 3.1 conditions on noisy latent variables $x^{(k)}$ and therefore requires a precise notion of the conditional distribution over executed actions given $(s, x^{(k)}, k)$. While a diffusion policy specifies how clean actions are generated through reverse-time denoising, it does not by itself define a joint distribution over actions and intermediate latent variables $(a, x^{(k)})$. To make conditioning on $x^{(k)}$ well-defined in a manner consistent with the environment —and to keep all value-related quantities defined through the executed clean actions $a \in \mathcal{A}$— we introduce a surrogate semantic model that links noisy latent variables to executable actions through the forward diffusion process.

**Definition 3.3** (Surrogate Noisy Joint). Let $\mu(a \mid s)$ denote a reference action distribution over $\mathcal{A}$ given a state $s$. Moreover, let $\mu(a \mid s)$ represent the marginal action distribution induced by the diffusion policy $\pi_\theta$ at state $s$. For each diffusion step $k > 0$, we define a surrogate joint distribution over actions and noisy latent variables by

$$\tilde{p}_k(a, x^{(k)} \mid s) \triangleq \mu(a \mid s)\, q_k(x^{(k)} \mid a), \quad (7)$$

where $q_k(x^{(k)} \mid a)$ is the forward diffusion kernel.

Intuitively, the surrogate joint in Definition 3.3 can be viewed as the distribution obtained by first sampling an executable action from the policy's own marginal action distribution and then corrupting it through the forward diffusion process to obtain a noisy latent variable. Under this

construction, conditioning on $x^{(k)}$ corresponds to reasoning about which clean actions, under the policy's induced action distribution, are compatible with the observed noisy latent state via the diffusion noising mechanism.

The surrogate joint in Definition 3.3 induces a conditional distribution $\tilde{p}_k(a \mid s, x^{(k)})$ that specifies which executable actions are compatible with a given noisy latent state under the diffusion corruption process. This conditional is used exclusively to define the gradients of the noisy-space value function. Importantly, the surrogate semantic model does not constrain the diffusion policy or alter its reverse-time dynamics; it serves only as a principled interface between value functions defined on executed actions and optimization carried out in noisy latent space.

### 3.3. Noisy-Space Policy Gradient (NSPG)

The noisy-space action-value function defined in Section 3.1 provides an objective for latent variable optimization in diffusion policies. Policy improvement therefore corresponds to ascending $Q_{\mathrm{noisy}}^{\pi_\theta}(s, x^{(k)}, k)$ with respect to the noisy latent variable $x^{(k)}$. Our goal in this section is to characterize this ascent in a form that depends only on local quantities at diffusion step $k$, enabling efficient optimization without backpropagating through the full denoising trajectory.

Using the surrogate semantics introduced in Section 3.2 to define conditioning on noisy latent variables, we can derive a closed-form expression for the gradient of the noisy-space action-value function.

**Theorem 3.4** (Noisy-Space Policy Gradient). *Let $\pi_\theta$ be a diffusion policy and let $Q_{\mathrm{noisy}}^{\pi_\theta}$ be defined as in Definition 3.1. For a fixed state $s$ and diffusion step $k$, consider the conditional distribution over executed actions induced by the policy when initialized at the noisy latent state $x^{(k)}$, with conditioning semantics defined as in Section 3.2. Then, the gradient of the noisy-space action-value function satisfies*

$$\nabla_{x^{(k)}} Q_{\mathrm{noisy}}^{\pi_\theta}(s, x^{(k)}, k) = \mathbb{E}_{a \sim \pi_\theta} \Big[$$
$$A_{\mathrm{noisy}}^{\pi_\theta}(s, a, x^{(k)}, k) \nabla_{x^{(k)}} \log q_k(x^{(k)} \mid a) \Big]. \quad (8)$$

*Here, $A_{\mathrm{noisy}}^{\pi_\theta}(s, a, x^{(k)}, k)$ denotes the noisy-space advantage:*

$$A_{\mathrm{noisy}}^{\pi_\theta}(s, a, x^{(k)}, k) = Q^{\pi_\theta}(s, a) - Q_{\mathrm{noisy}}^{\pi_\theta}(s, x^{(k)}, k). \quad (9)$$

*Proof.* The proof can be found in the appendix. $\square$

We estimate $Q^{\pi_\theta}$ and (8) by Monte Carlo sampling: draw $a_i \sim \pi_\theta$ and use $\widehat{Q}_{\mathrm{noisy}}^{\pi_\theta} = \frac{1}{N} \sum_{i=1}^{N} Q^{\pi_\theta}(s, a_i)$, plugging $\widehat{A}_{\mathrm{noisy}}^{\pi_\theta}(s, a_i, x^{(k)}, k) = Q^{\pi_\theta}(s, a_i) - \widehat{Q}_{\mathrm{noisy}}^{\pi_\theta}$ into the sample average of the score-weighted gradient. We further study the impact of this Monte Carlo estimation procedure in the

experimental section, where we ablate the number of action samples used to estimate $Q_{\mathrm{noisy}}^{\pi_\theta}$ and analyze its effect on stability and performance.

---

**Interpretation of Noisy-Space Policy Gradient**

The expression in (8) gives the gradient of the noisy-space value function. The score term $\nabla_{x^{(k)}} \log q_k(x^{(k)} \mid a)$ describes how infinitesimal perturbations of the noisy latent variable affect its likelihood of originating from action $a$ under the forward diffusion process. The noisy-space advantage $A_{\mathrm{noisy}}^{\pi_\theta}(s, a, x^{(k)}, k)$ modulates this score, reinforcing directions in noisy space associated with above-average actions while suppressing those associated with below-average ones.

Notably, this baseline arises directly from marginalization and is not introduced as a heuristic variance-reduction device.

---

This perspective clarifies why NSPG can differ from naive decoded-action gradients, especially in multimodal action distributions where the most likely action region need not be the one associated with the highest reward. A naive decoded-action gradient first maps each noisy latent $x^{(k)}$ to a clean action estimate $\hat{a}(x^{(k)})$ and then differentiates $Q(s, \hat{a}(x^{(k)}))$. This is similar in spirit to decoded-action value guidance used in prior diffusion RL methods such as Diffusion Q-Learning (Wang et al., 2022). However, under the diffusion process, a noisy latent generally corresponds to a distribution of compatible clean actions rather than a single action. NSPG instead differentiates the expected clean-action value under the diffusion-induced distribution, $Q_{\mathrm{noisy}}$, so directions in noisy space are reinforced according to the relative value of the clean actions that could have generated the latent. Consequently, high-value compatible actions exert positive influence through the score term, while below-average actions are suppressed by the noisy-space advantage. Figure 4 illustrates this distinction in a two-dimensional toy example with two action regions: one that is more likely under the behavior prior but has lower reward, and another that is less likely but has higher reward. The illustration shows that NSPG induces a stronger directional bias toward the high-reward region in this setting, whereas the naive decoded-action gradient follows the attraction basins induced by the chosen clean-action estimate. Full details are in Appendix B.6.

**Relation to Advantage-Weighted Regression.** At a high level, the noisy-space policy gradient in (8) may, at first, bear resemblance to advantage-weighted regression (AWR) (Peng et al., 2019), in that both incorporate an advantage signal to modulate updates. However, the two approaches differ fundamentally. AWR uses the advantage to reweight

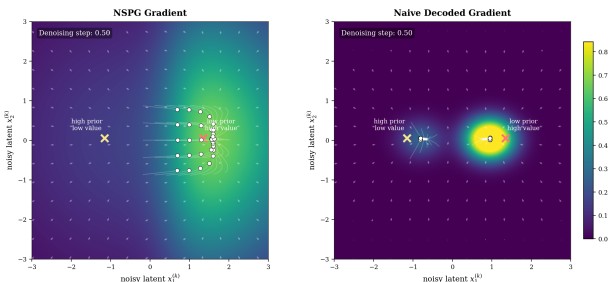

*Figure 1.* **Toy illustration of noisy-space gradients.** We consider a two-dimensional multimodal action distribution with a high-density low-reward region and a lower-density high-reward region. NSPG follows the gradient of $Q_{\text{noisy}}$, the expected clean-action value under the diffusion-induced distribution, which induces guidance toward the higher-reward region. In contrast, the decoded gradient differentiates $Q(s, \hat{a}(x^{(k)}))$ after mapping each noisy latent to a single clean action. Arrows show the corresponding noisy-space gradient fields, and particle traces illustrate gradient-ascent rollouts across denoising steps.

state–action pairs drawn from a fixed dataset within a supervised regression objective in action space. In contrast, our method does not reweight data samples. Instead, it samples actions from the policy $a \sim \pi_\theta$ and uses the noisy-space advantage $A^{\pi_\theta}_{\text{noisy}}$ as a signed modulator of the score term $\nabla_{x^{(k)}} \log q_k(x^{(k)} \mid a)$, thereby amplifying or suppressing gradient contributions from different sampled actions. Crucially, this advantage term arises naturally from marginalization when differentiating the noisy-space action-value function, yielding a true value-function gradient rather than a reweighted imitation objective. In the following section, we show how this gradient identity leads directly to a practical regression objective for training diffusion policies.

# 4. Learning Diffusion Policies with Noisy-Space Actor–Critic (NSAC)

In this section, we translate the noisy-space policy gradient framework developed in Section 3.3 into a practical offline RL algorithm for diffusion policies. We introduce *Noisy-Space Actor–Critic (NSAC)*, an actor–critic method that performs policy improvement directly in noisy latent space while maintaining value estimation strictly in action space. NSAC alternates between (i) improving a diffusion-based policy via noisy-space policy gradients and (ii) learning a conservative action-space critic from offline data.

## 4.1. Actor: Noisy-Space Policy Improvement

We begin from behavior-constrained policy iteration(Eq. 1), which admits a closed-form solution for the optimal policy under a KL constraint:

$$\pi^*(a \mid s) \propto \pi_\beta(a \mid s) \exp\left(\tfrac{1}{\eta} Q^\pi(s, a)\right), \quad (10)$$

where $\pi_\beta$ is the behavior policy and $\eta$ is a temperature parameter. Directly optimizing (10) is infeasible for diffusion policies due to the intractability of policy densities. This difficulty can be avoided by rewriting the policy improvement objective in terms of score functions:

$$\nabla_a \log \pi^*(a \mid s) = \nabla_a \log \pi_\beta(a \mid s) + \tfrac{1}{\eta} \nabla_a Q^\pi(s, a). \quad (11)$$

To accommodate diffusion policies, we extend the score-based policy improvement objective from action space to noisy perturbations induced by the forward diffusion process. Following explicit score matching for denoising models (Vincent, 2011), this yields the following noisy-space score relation:

$$\nabla_{x^{(k)}} \log p^*(x^{(k)} \mid s) = \nabla_{x^{(k)}} \log p_k(x^{(k)} \mid s) + \tfrac{1}{\eta} \nabla_{x^{(k)}} Q_{noisy}(s, x^{(k)}, k), \quad (12)$$

We note that this formulation differs from regular diffusion actor-critic methods (Fang et al., 2024) in the value term: instead of applying an action-value function directly to noisy latents, NSAC uses $Q_{noisy}$. Since $Q_{noisy}$ coincides with the expected action-space return under the diffusion-induced action distribution (proposition 3.2), replacing the value term in the score-based policy improvement objective preserves the score-function structure.

**Actor objective.** Following the same score-matching derivation as in prior diffusion actor–critic methods (Fang et al., 2024), the noisy-space policy improvement objective admits a diffusion-compatible regression form. Rearranging terms and absorbing constants into the temperature parameter yields the final actor loss:

$$\mathcal{L}_{actor}(\theta) = \mathbb{E}_{(s,a)\sim\mathcal{D}, k, \epsilon}\Big[\eta \big\|\epsilon_\theta(x^{(k)}, s, k) - \epsilon\big\|^2 +$$
$$\lambda \sqrt{1 - \bar{\alpha}_k}\, \epsilon_\theta(x^{(k)}, s, k)^\top \nabla_{x^{(k)}} Q_{noisy}(s, x^{(k)}, k)\Big], \quad (13)$$

where $x^{(k)} = \sqrt{\bar{\alpha}_k} a + \sqrt{1 - \bar{\alpha}_k}\epsilon$. Here, $\lambda$ is a guidance strength parameter that controls the influence of value-based guidance relative to the denoising objective.

## 4.2. Critic: Pessimistic Action-Space Value Learning

The critic in NSAC is an action-space Q-function $Q_\phi(s, a)$ trained on offline data using a pessimistic value estimation objective (Ghasemipour et al., 2022). This design mitigates overestimation and distributional shift by penalizing actions with limited support under the offline data distribution. More specifically, the critic is trained via Bellman regression with a lower-confidence bound (LCB) target:

$$\mathcal{L}_{critic}(\phi) = \mathbb{E}_{(s,a,r,s')\sim\mathcal{D}, a'\sim\pi_\theta}\Big[r + \gamma Q_{LCB}(s', a')$$
$$- Q_\phi(s, a)\Big]^2, \quad (14)$$

**Algorithm 1** Noisy-Space Actor–Critic (NSAC)

---

**Require:** Offline dataset $\mathcal{D}$; diffusion actor $\pi_\theta$; critic ensemble $\{Q_{\phi_h}\}_{h=1}^H$; guidance strength $\lambda$; temperature $\eta$; pessimism coefficient $\rho$.

1: Initialize actor parameters $\theta$ and critic parameters $\{\phi_h\}_{h=1}^H$.
2: **while** not converged **do**
3:     Sample a minibatch of transitions $(s, a, r, s') \sim \mathcal{D}$.
4:     Sample diffusion step $k$ and noise $\epsilon$; Construct noisy action $x^{(k)}$ via the forward diffusion process $q(x^{(k)}|a)$.
5:     Sample next action $a' \sim \pi_\theta(\cdot \mid s')$.
6:     Update the critic using Eq. 14.
7:     Compute $\nabla_{x^{(k)}} Q_{noisy}(s, x^{(k)}, k)$ using (Eqs. 8, 9).
8:     Update the diffusion actor using Eq. (13).
9: **end while**
10: **return** diffusion policy $\pi_\theta$.

---

where the LCB value is defined as

$$Q_{LCB}(s', a') = \mathbb{E}_h[Q_{\phi_h}(s', a')] - \rho \sqrt{\mathrm{Var}_h(Q_{\phi_h}(s', a'))}. \tag{15}$$

Here, $h$ indexes the critic ensemble with parameters $\{\phi_h\}$; $\mathbb{E}_h[\cdot]$ and $\mathrm{Var}_h(\cdot)$ denote expectation and variance over the ensemble, and $\rho > 0$ controls the pessimism level.

During actor optimization, the action-space critic $Q_\phi(s, a)$ is used solely to construct the noisy-space action-value function $Q_{\mathrm{noisy}}(s, x^{(k)}, k)$ via conditional expectation. In this phase, the critic is evaluated only on executable actions $a \in \mathcal{A}$ and is never evaluated on noisy latent variables. More generally, the critic's role is to estimate the action-value function of the underlying MDP in action space, preserving a clear separation between conservative value estimation and latent-space policy optimization.

### 4.2.1. ALGORITHM SUMMARY

Algorithm 1 summarizes NSAC. The method alternates between learning a pessimistic action-space critic using the lower-confidence bound objective in (14)–(15) and improving a diffusion actor via the noisy-space policy improvement objective in (13). Importantly, the critic is never evaluated on noisy latents; all value information enters the actor only through the noisy-space value $Q_{noisy}$ and its gradient.

## 5. Prior Work

**Offline Reinforcement Learning.** Offline reinforcement learning seeks to learn effective policies from fixed datasets without additional environment interaction (Levine et al., 2020). Prior work has largely focused on value-based methods augmented with regularization to mitigate distributional shift. Policy-regularization approaches constrain policy improvement toward the behavior policy, including methods

based on conditional generative models such as BCQ (Fujimoto et al., 2019), divergence-based regularization such as BEAR (Kumar et al., 2019), and KL-regularized actor–critic methods such as BRAC (Wu et al., 2019) and TD3+BC (Fujimoto & Gu, 2021).

Complementary work addresses overestimation on out-of-distribution actions through pessimistic or conservative value estimation. Representative methods include CQL (Kumar et al., 2020), IQL (Kostrikov et al., 2021), IDQL (Hansen-Estruch et al., 2023), and IVR (Xu et al., 2023), which employ conservative objectives, asymmetric losses, or in-sample maximization to obtain reliable value estimates. Related approaches include Extreme Q-learning (Garg et al., 2023), which estimates maximal Q-values via Gumbel regression, as well as methods that mitigate distributional shift through explicit out-of-distribution detection (Yu et al., 2020; Kidambi et al., 2020; An et al., 2021; Nikulin et al., 2023) or dual optimization formulations (Lee et al., 2021; Sikchi et al., 2023).

Beyond target policy regularization, recent work emphasizes constraining the *policy improvement step* itself to ensure stable learning (Zhuang et al., 2023; Zhang & Tan, 2024; Nair et al., 2020; Peng et al., 2019). Trust-region and KL-regularized methods (Schulman et al., 2015; 2017) explicitly limit divergence between successive policies, mitigating error amplification from bootstrapped value estimates. Our approach follows this line of work by adopting KL-regularized policy iteration, which enables controlled updates while remaining compatible with expressive generative policies.

**Diffusion and Flow Policies for Offline Reinforcement Learning.** Recent work has explored diffusion and flow models as expressive policy parameterizations for offline RL due to their ability to represent complex, multimodal action distributions. A first class of methods uses diffusion models primarily as *behavior samplers* or planners rather than optimizing them directly with RL objectives. For example, Diffuser (Janner et al., 2022) models trajectories for return-conditioned planning via guided sampling, while Diffusion Q-learning (Wang et al., 2022), IDQL (Hansen-Estruch et al., 2023), SfBC (Chen et al., 2022), and Diffusion trusted Q-learning (Chen et al., 2024) generate candidate actions or behavior policies from which target policies are extracted. While these approaches avoid assigning value to noisy latent variables, they typically rely on repeated sampling and value evaluation, limiting scalability.

A second line of work directly optimizes diffusion- or flow-based policies with value-based objectives. Methods such as DiffCPS (He et al., 2023), Consistency-AC (Ding & Jin, 2023), SRDP (Ada et al., 2024), and (Zhang et al., 2025) use reparameterized policy gradients, differentiating action-space value functions through samples from the generative

policy, typically with diffusion or flow regularization. While straightforward, these methods often require backpropagation through the generative process, adding computational overhead and potential instability. Other approaches improve tractability by changing the policy class. For example, FQL (Park et al., 2025) uses a one-step flow policy, avoiding iterative denoising and simplifying optimization, but trading off part of the expressive capacity that motivates multi-step diffusion policies. ReinFlow (Zhang et al., 2026) offers a complementary online RL perspective, injecting noise into flow policies to obtain a tractable Markov formulation and exact likelihoods for policy-gradient optimization. BDPO (Gao et al., 2025) also introduces value functions at intermediate diffusion steps; however, these values estimate cumulative KL penalties along denoising, rather than environment returns. DAC (Fang et al., 2024) is closest to our work: it enables score-function-based policy improvement without backpropagating through denoising, but applies clean action-value functions directly to noisy latents. In contrast, our approach optimizes noisy latents using updates computed from clean action-space value estimates, while retaining the expressivity of multi-step diffusion policies.

## 6. Experiments

In this section, we empirically evaluate NSAC to assess the effectiveness of noisy-space policy optimization for diffusion policies in offline RL. Our experiments are designed to answer three core questions: (i) how NSAC performs against established offline RL baselines across state-based and vision-based benchmarks; (ii) whether the proposed noisy-space action-value formulation, which defines $Q_{\mathrm{noisy}}$ through expected clean action-space returns, improves over diffusion actor–critic methods such as DAC (Fang et al., 2024) that apply action-value functions directly to noisy latents; and (iii) how sensitive NSAC is to key design choices, including guidance strength, pessimism strength, critic ensemble size, and the Monte Carlo approximation of noisy-space expectations.

Full experimental details, including environments, architectures, hyperparameters, implementation choices, and ablation studies on the effect of critic ensembles as well as the pessimism factor, are provided in Appendix C.

**Baselines.** We compare NSAC against an extensive collection of representative offline RL baselines that span multiple algorithmic paradigms. For explicit policy-regularization methods, we include Behavioral cloning (BC) One-Step RL (Onestep-RL) (Brandfonbrener et al., 2021), which performs a single-step actor–critic update, as well as Conservative Q-learning (CQL) (Kumar et al., 2020) and Implicit Q-learning (IQL) (Kostrikov et al., 2021), which constrain policy learning through conservative value estimation and in-sample

maximization, respectively. We further compare against Implicit Value Regularization (IVR) (Brandfonbrener et al., 2021) and Extreme Q-learning (XQL) (Garg et al., 2023), which approximate maximal Q-values using asymmetric objectives or Gumbel-based regression.

In addition to Gaussian-policy baselines, we compare against methods that use expressive generative policies, including diffusion- and flow-based models. Some use diffusion models as *behavior samplers* or planners: for example, Diffuser (Janner et al., 2022) trains a diffusion model over trajectories and performs return-conditioned planning via guided sampling, while Diffusion Q-learning (Wang et al., 2022) treats a diffusion model as a biased behavior sampler and augments training with an auxiliary loss that encourages denoised actions to achieve high Q-values. Other approaches use diffusion models to model the behavior policy itself (e.g., IDQL (Hansen-Estruch et al., 2023)) and SfBC (Chen et al., 2022)), from which target policies are obtained by resampling or by extracting simpler parametric policies. Diffusion Trusted Q-learning, or DTQL (Chen et al., 2024) likewise models behavior samplers with diffusion models and then derives policies focusing on high-reward modes. FQL (Park et al., 2025) provides a flow-based counterpart, replacing multi-step diffusion policies with one-step flows to simplify optimization.

We also include Diffusion Actor–Critic (DAC) (Fang et al., 2024) as the closest score-based diffusion actor–critic baseline. DAC avoids backpropagation through the denoising process, but applies action-value functions directly to noisy latents. In contrast, NSAC uses $Q_{\mathrm{noisy}}$ to relate noisy latents to expected clean action-space returns, making DAC a targeted baseline for isolating the effect of our noisy-space action-value formulation.

For the OGBench visual environments, we use a targeted baseline set following recent flow-policy evaluations (Park et al., 2025), focusing on strong baselines for high-dimensional visual offline RL under a tractable evaluation budget. Besides IQL and FQL, we include Re-BRAC (Tarasov et al., 2023), a strong minimalist Gaussian-policy baseline, as well as FBRAC and IFQL (Park et al., 2025). FBRAC is the flow counterpart of DQL (Wang et al., 2022), while IFQL is the flow counterpart of IDQL (Hansen-Estruch et al., 2023). We also report DAC on these tasks to provide the same targeted comparison in the visual setting.

**Main results.** Table 1 reports average normalized scores for NSAC across a diverse set of offline RL benchmarks, including locomotion, AntMaze, and Adroit tasks. Overall, NSAC demonstrates strong and consistent performance, outperforming or matching prior methods across different environments. On standard locomotion benchmarks, NSAC performs competitively across medium, replay, and

*Table 1.* Average normalized scores of NSAC over four runs compared with baseline methods (mean ± standard deviation). Best results, as well as results within 5% of the best performance for each task, are highlighted .

| Dataset | BC | Onestep-RL | CQL | IQL | IVR | XQL | Diffuser | DTQL | AlignIQL | SfBC | DQL | IDQL-A | FQL[0] | DAC | NSAC (ours) |
|---|---|---|---|---|---|---|---|---|---|---|---|---|---|---|---|
| HalfCheetah-medium-v2 | 42.6 | 48.4 | 44.0 | 47.4 | 48.3 | 48.3 | 44.2 | 57.9 | 46.0 | 45.9 | 51.1 | 51.0 | 60.1 | 59.1 | 64.9±0.6 |
| Hopper-medium-v2 | 52.9 | 59.6 | 58.5 | 66.3 | 75.5 | 74.2 | 58.5 | 99.6 | 56.1 | 57.1 | 90.5 | 65.4 | 74.5 | 101.2 | 103.7±0.9 |
| Walker2d-medium-v2 | 75.6 | 81.8 | 72.5 | 78.3 | 84.2 | 84.2 | 79.7 | 89.4 | 78.5 | 77.9 | 87.0 | 82.5 | 72.7 | 96.8 | 97.6±0.8 |
| HalfCheetah-medium-replay-v2 | 36.3 | 38.1 | 45.5 | 44.2 | 44.8 | 45.2 | 42.2 | 50.9 | 41.1 | 37.1 | 47.8 | 45.9 | 51.1 | 55.0 | 59.4±0.6 |
| Hopper-medium-replay-v2 | 18.1 | 97.5 | 95.0 | 94.7 | 99.7 | 100.7 | 96.8 | 100.0 | 74.8 | 86.2 | 101.3 | 92.1 | 85.4 | 103.1 | 104.5±0.7 |
| Walker2d-medium-replay-v2 | 26.0 | 49.5 | 77.2 | 73.9 | 81.2 | 82.2 | 61.2 | 88.5 | 76.5 | 65.1 | 95.5 | 85.1 | 82.1 | 96.8 | 98±1.5 |
| HalfCheetah-medium-expert-v2 | 55.2 | 93.4 | 91.6 | 86.7 | 94.0 | 94.2 | 79.8 | 92.7 | 89.1 | 92.6 | 96.8 | 95.9 | 99.8 | 99.1 | 104.4±0.8 |
| Hopper-medium-expert-v2 | 52.5 | 103.3 | 105.4 | 91.5 | 111.8 | 111.2 | 107.2 | 109.3 | 107.1 | 108.6 | 111.1 | 108.6 | 86.2 | 111.7 | 113±0.8 |
| Walker2d-medium-expert-v2 | 101.9 | 113.0 | 108.8 | 109.6 | 110.2 | 112.7 | 108.4 | 110.0 | 111.9 | 109.8 | 110.1 | 112.7 | 100.5 | 113.6 | 116.1±0.6 |
| Locomotion total | 461.1 | 684.6 | 698.5 | 692.6 | 749.7 | 752.9 | 678.0 | 798.3 | 681.1 | 680.3 | 791.2 | 739.2 | 712.4 | 836.4 | 861.7 |
| AntMaze-medium-play-v0 | 0.0 | 0.3 | 61.2 | 71.2 | 80.2 | 76.0 | – | 79.6 | 80.5 | 81.3 | 76.6 | 84.2 | 72 | 85.8 | 92.4±2.1 |
| AntMaze-medium-diverse-v0 | 0.0 | 0.0 | 53.7 | 70.0 | 79.1 | 73.6 | – | 82.2 | 85.5 | 82.0 | 78.6 | 84.8 | 68 | 84.0 | 89±2.9 |
| AntMaze-large-play-v0 | 0.0 | 0.0 | 15.8 | 39.6 | 53.2 | 46.5 | – | 52.0 | 65.2 | 59.3 | 46.4 | 63.5 | 42 | 50.3 | 77.8±3.8 |
| AntMaze-large-diverse-v0 | 0.0 | 0.0 | 14.9 | 47.5 | 52.3 | 49.0 | – | 66.4 | 54.0 | 45.5 | 56.6 | 67.9 | 48 | 55.3 | 78±3.0 |
| AntMaze total | 0.0 | 0.3 | 145.6 | 228.3 | 264.8 | 245.1 | – | 280.2 | 285.2 | 268.1 | 258.2 | 300.4 | 230 | 275.4 | 337.2 |
| Pen-Human-v1 | 25.8 | 68.9 | 35.2 | 71.5 | – | – | – | 64.1 | – | – | 72.8 | – | 53 | 81.3 | 83.9±3.4 |
| Pen-Cloned-v1 | 38.3 | 44.0 | 27.2 | 37.3 | – | – | – | 81.3 | – | – | 57.3 | – | 74 | 63.9 | 84.6±4.0 |
| Adroit total | 64.1 | 112.9 | 62.4 | 108.8 | – | – | – | 145.4 | – | – | 130.1 | – | 127 | 145.2 | 168.5 |

expert datasets, maintaining strong performance in well-behaved continuous control settings. Notably, on HalfCheetah medium and medium-replay tasks—characterized by long-horizon dynamics and sensitivity to action perturbations—NSAC achieves gains of approximately 10% over prior methods.

On AntMaze tasks, which are characterized by sparse rewards and datasets containing numerous suboptimal trajectories, NSAC yields substantial improvements over existing methods. In particular, NSAC significantly outperforms DAC on large maze environments, achieving improvements of over 50% relative performance on `AntMaze-large-play` and `AntMaze-large-diverse`. These results highlight the importance of defining value guidance through executable actions when optimizing diffusion policies in long-horizon, multimodal settings.

We observe similar trends on high-dimensional dexterous manipulation tasks. On `Pen-Cloned-v1`, NSAC achieves the strongest performance among all evaluated methods, demonstrating that noisy-space value guidance remains effective in settings with complex action spaces and limited data coverage.

Table 2 further evaluates NSAC on vision-based OGBench tasks, where policies must operate from high-dimensional visual observations. NSAC achieves the best or near-best performance on all four tasks and obtains the highest aggregate score, improving over the strongest prior baseline from 253.2 to 271.5. The gains are particularly clear on the visual cube tasks and the more challenging 4x4 puzzle task, where NSAC outperforms both flow-policy baselines and

DAC. These results indicate that the proposed noisy-space action-value formulation remains effective beyond state-based control, extending to visual offline RL settings where representation learning, multimodal action distributions, and value-based policy improvement are jointly challenging.

Taken together, these results demonstrate that NSAC provides a robust and scalable approach for diffusion-based offline RL, with particularly pronounced benefits in challenging environments where value estimation and policy optimization are tightly coupled.

### 6.1. Ablation Studies

We conduct targeted ablation studies to isolate the effects of key components in NSAC. Specifically, we examine design choices that are central to noisy-space policy optimization: (1) the strength of value guidance in the diffusion actor and (2) the approximation of the noisy-space action-value expectation. In terms of computational complexity, NSAC remains efficient with a small number of Monte Carlo samples: with two samples, NSAC is approximately 10% faster than prior diffusion actor–critic methods (Fang et al., 2024), while with four samples the training time is comparable. Additional ablations on critic pessimism, ensemble size, and a detailed runtime analysis are provided in the appendix.

**Effect of Guidance Strength.** We analyze the effect of the guidance strength $\lambda$, which balances noisy-space value guidance against the behavior cloning objective. As shown

---

[0]Some baseline scores are reported in prior work on different benchmark versions; e.g., FQL reports AntMaze on v2, whereas we use v0 following broader offline RL evaluation practice.

*Table 2.* Offline RL results on OGBench visual environments, reported as success rates (mean ± standard deviation). Best results, as well as results within 5% of the best performance for each task, are highlighted .

| Task | IQL | ReBRAC | FBRAC | IFQL | FQL | DAC | NSAC (ours) |
|---|---|---|---|---|---|---|---|
| visual-cube-single-play-singletask-task1-v0 | 70 ±12 | 83 ±6 | 55 ±8 | 49 ±7 | 81 ±12 | 83.7 ±10 | 88±7 |
| visual-cube-double-play-singletask-task1-v0 | 34 ±23 | 4 ±4 | 6 ±2 | 8 ±6 | 21 ±11 | 34 ±12 | 36 ±14 |
| visual-puzzle-3x3-play-singletask-task1-v0 | 7 ±15 | 88 ±4 | 7 ±2 | 100 ±0 | 94 ±1 | 98.2 ±2 | 100 ±0 |
| visual-puzzle-4x4-play-singletask-task1-v0 | 0 ±0 | 26 ±6 | 0 ±0 | 8 ±15 | 33 ±6 | 40 ±5.7 | 47.5 ±4 |
| OGBench total | 111 | 201 | 68 | 165 | 229 | 255.9 | 271.5 |

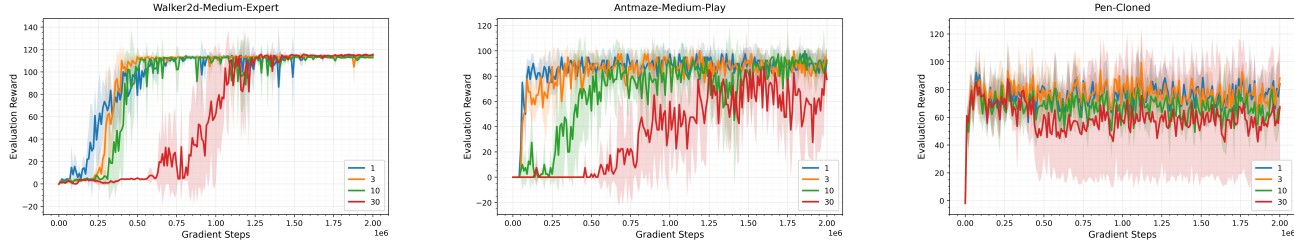

*Figure 2.* Effect of guidance strength on learning dynamics across different tasks. NSAC is robust across a range of guidance strengths, but overly aggressive guidance can degrade performance.

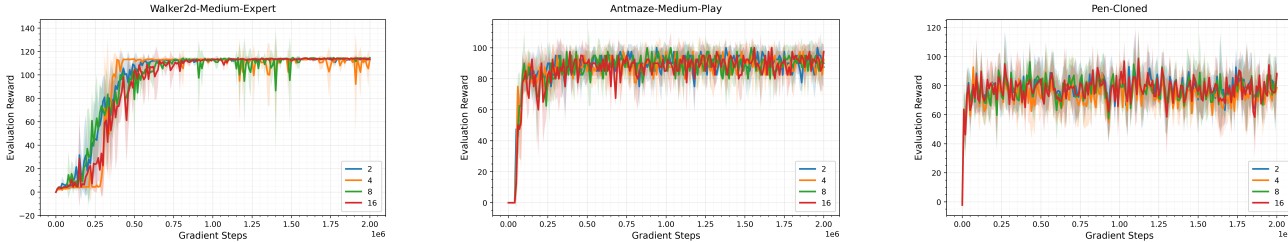

*Figure 3.* Effect of the number of Monte Carlo samples used to approximate the noisy-space expectation on learning dynamics. NSAC remains stable with few samples, indicating that the noisy-space expectation can be estimated efficiently.

in Fig. 2, NSAC exhibits stable learning behavior across a wide range of $\lambda$, with intermediate values ($\lambda \in \{1, 3, 10\}$) yielding the strongest performance. In contrast, excessively strong guidance degrades learning. This behavior is consistent with the challenges of offline RL: overly aggressive value optimization can drive the policy away from the support of the offline dataset, increasing distributional shift and exposing the policy to unreliable value estimates. Moderate guidance therefore achieves a better balance between policy improvement and behavioral regularization.

**Approximating the Noisy-Space Expectation.** NSAC approximates the noisy-space action-value function $Q_{\text{noisy}}$ via Monte Carlo estimation over actions induced by the diffusion policy. We analyze sensitivity to the number of samples used in this approximation. As shown in Fig. 3, NSAC achieves strong performance with only a small number of samples, while increasing the sample count yields diminishing returns. These results indicate that the noisy-space expectation can be estimated efficiently in practice, without relying on expensive sampling procedures.

## 7. Discussion and Conclusion

In this work, we introduced NSPG, a principled framework for optimizing diffusion policies in offline RL through value functions defined over noisy latent variables. By assigning values to noisy latents through the distribution of clean actions they induce, NSPG optimizes expected action-space returns rather than relying on a single decoded action. This yields a principled noisy-space objective while querying the critic only on clean, executable actions rather than noisy latents. Building on this formulation, NSAC achieves strong empirical performance across state-based and vision-based benchmarks, with especially clear gains in sparse-reward, long-horizon, and multimodal settings.

Beyond offline RL, the NSPG perspective may extend naturally to online and offline-to-online learning, where diffusion policies can continue to benefit from expressive generative representations. We leave a systematic investigation of these directions to future work, and hope that this framework informs the design of value-based objectives for broader classes of diffusion- and flow-based policy models.

## Acknowledgements

This work was supported by Vinnova, Sweden, through the AllDrive project. The computations and data handling were enabled by resources provided by the National Academic Infrastructure for Supercomputing in Sweden (NAISS), partially funded by the Swedish Research Council through grant agreement no. 2022-06725.

## Impact Statement

This work aims to advance reinforcement learning by improving the training of diffusion policies in offline settings. The proposed methods are evaluated on standard simulated benchmarks and are intended to contribute to more reliable and data-efficient policy learning. As with other offline RL methods, potential societal impacts depend on the downstream applications in which such policies are deployed. We do not identify any direct negative societal consequences that require specific discussion beyond the broader considerations associated with reinforcement learning and autonomous decision-making systems.

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

# A. Proofs

## A.1. Proof of Proposition 3.2

We restate Proposition 3.2. Let $\pi_\theta$ be a diffusion policy and let

$$Q^{\pi_\theta}_{\text{noisy}}(s, x_t, t) \triangleq \mathbb{E}_{a \sim \pi_\theta(\cdot | s, x_t, t)}[Q^{\pi_\theta}(s, a)]$$

Under the assumption that the environment transition kernel $P(\cdot \mid s, a)$ and reward function $r(s, a)$ depend only on the executed action $a$ (and do not depend directly on $x_t$), we show that

$$Q^{\pi_\theta}_{\text{noisy}}(s, x_t, t) = \mathbb{E}\left[\sum_{k=0}^{\infty} \gamma^k r(s_k, a_k) \,\middle|\, s_0 = s, \ x_t\right],$$

where $a_0 \sim \pi_\theta(\cdot \mid s_0, x_t, t)$ and, for $k \geq 1$, $a_k$ is sampled from the marginal action distribution induced by $\pi_\theta$ given the current state $s_k$.

**Proof.** Fix any $(s, x_t, t)$. Consider the trajectory distribution generated as follows: set $s_0 = s$; sample the first action

$$a_0 \sim \pi_\theta(\cdot \mid s_0, x_t, t);$$

then sample the next state $s_1 \sim P(\cdot \mid s_0, a_0)$; and for all $k \geq 1$, sample actions $a_k$ according to the (state-conditional) marginal action distribution of $\pi_\theta$ and transitions $s_{k+1} \sim P(\cdot \mid s_k, a_k)$.

By the assumed environment interface, conditioning on $(s_0 = s, x_t)$ affects the environment only through the distribution of the executed action $a_0$; once $a_0$ is fixed, the subsequent evolution of $(s_k, a_k)_{k \geq 1}$ depends on the policy only through its (state-conditional) marginal behavior and does not depend on $x_t$ directly. In particular, conditioning additionally on $a_0$ yields

$$\mathbb{E}\left[\sum_{k=0}^{\infty} \gamma^k r(s_k, a_k) \,\middle|\, s_0 = s, \ x_t, \ a_0\right] = Q^{\pi_\theta}(s, a_0),$$

by the definition of the standard action-value function $Q^{\pi_\theta}(s, a)$.

Taking expectation over $a_0 \sim \pi_\theta(\cdot \mid s, x_t, t)$ and applying the law of total expectation gives

$$\mathbb{E}\left[\sum_{k=0}^{\infty} \gamma^k r(s_k, a_k) \,\middle|\, s_0 = s, \ x_t\right] = \mathbb{E}_{a_0 \sim \pi_\theta(\cdot | s, x_t, t)}\left[\mathbb{E}\left[\sum_{k=0}^{\infty} \gamma^k r(s_k, a_k) \,\middle|\, s_0 = s, \ x_t, \ a_0\right]\right]$$

$$= \mathbb{E}_{a_0 \sim \pi_\theta(\cdot | s, x_t, t)}[Q^{\pi_\theta}(s, a_0)]$$

$$= Q^{\pi_\theta}_{\text{noisy}}(s, x_t, t),$$

which proves the proposition. $\qquad\square$

## A.2. Proof of Theorem 3.4

Fix $s \in \mathcal{S}$, diffusion step $k \in \{0, \ldots, T\}$, and $x^{(k)} \in \mathbb{R}^{d_a}$. For brevity, write

$$\pi_k(a) \triangleq \pi_\theta(a \mid s, x^{(k)}, k), \qquad Q(a) \triangleq Q^{\pi_\theta}(s, a), \qquad Q_{\text{noisy}} \triangleq Q^{\pi_\theta}_{\text{noisy}}(s, x^{(k)}, k).$$

Assume $Q(\cdot)$ is integrable under $\pi_k(\cdot)$ and that differentiation under the integral sign is valid.

By Definition 3.1,

$$Q_{\text{noisy}} = \int_{\mathcal{A}} Q(a)\, \pi_k(a)\, da. \tag{16}$$

Differentiating (16) with respect to $x^{(k)}$ and using $\nabla\pi = \pi\nabla\log\pi$ yields

$$\nabla_{x^{(k)}} Q_{\text{noisy}} = \int_{\mathcal{A}} Q(a)\, \nabla_{x^{(k)}} \pi_k(a)\, da = \int_{\mathcal{A}} Q(a)\, \pi_k(a)\, \nabla_{x^{(k)}} \log \pi_k(a)\, da$$

$$= \mathbb{E}_{a \sim \pi_\theta(\cdot | s, x^{(k)}, k)}\left[Q^{\pi_\theta}(s, a)\, \nabla_{x^{(k)}} \log \pi_\theta(a \mid s, x^{(k)}, k)\right]. \tag{17}$$

We now invoke the surrogate semantics from Section 3.2, which specifies the conditional distribution over executable actions given $(s, x^{(k)}, k)$ via the forward diffusion kernel. In particular,

$$\pi_\theta(a \mid s, x^{(k)}, k) = \frac{\mu(a \mid s)\, q_k(x^{(k)} \mid a)}{\tilde{p}_k(x^{(k)} \mid s)}, \qquad \tilde{p}_k(x^{(k)} \mid s) \triangleq \int_{\mathcal{A}} \mu(a' \mid s)\, q_k(x^{(k)} \mid a')\, da', \tag{18}$$

where $\mu(a \mid s)$ is the reference action distribution and $q_k(x^{(k)} \mid a)$ is the forward diffusion kernel. Taking logs and differentiating with respect to $x^{(k)}$ (noting $\mu(\cdot \mid s)$ is independent of $x^{(k)}$) gives

$$\nabla_{x^{(k)}} \log \pi_\theta(a \mid s, x^{(k)}, k) = \nabla_{x^{(k)}} \log q_k(x^{(k)} \mid a) - \nabla_{x^{(k)}} \log \tilde{p}_k(x^{(k)} \mid s). \tag{19}$$

Substituting (19) into (17) yields

$$\nabla_{x^{(k)}} Q_{\text{noisy}} = \mathbb{E}_{a \sim \pi_\theta(\cdot \mid s, x^{(k)}, k)} \left[ Q^{\pi_\theta}(s, a)\, \nabla_{x^{(k)}} \log q_k(x^{(k)} \mid a) \right] - Q_{\text{noisy}}\, \nabla_{x^{(k)}} \log \tilde{p}_k(x^{(k)} \mid s). \tag{20}$$

It remains to express $\nabla_{x^{(k)}} \log \tilde{p}_k(x^{(k)} \mid s)$ as an expectation under the same conditional. Differentiating the marginal likelihood in (18) and rearranging gives

$$\begin{aligned}
\nabla_{x^{(k)}} \log \tilde{p}_k(x^{(k)} \mid s) &= \frac{1}{\tilde{p}_k(x^{(k)} \mid s)} \int_{\mathcal{A}} \mu(a \mid s)\, \nabla_{x^{(k)}} q_k(x^{(k)} \mid a)\, da \\
&= \frac{1}{\tilde{p}_k(x^{(k)} \mid s)} \int_{\mathcal{A}} \mu(a \mid s)\, q_k(x^{(k)} \mid a)\, \nabla_{x^{(k)}} \log q_k(x^{(k)} \mid a)\, da \\
&= \int_{\mathcal{A}} \frac{\mu(a \mid s)\, q_k(x^{(k)} \mid a)}{\tilde{p}_k(x^{(k)} \mid s)}\, \nabla_{x^{(k)}} \log q_k(x^{(k)} \mid a)\, da \\
&= \mathbb{E}_{a \sim \pi_\theta(\cdot \mid s, x^{(k)}, k)} \left[ \nabla_{x^{(k)}} \log q_k(x^{(k)} \mid a) \right],
\end{aligned} \tag{21}$$

where the final equality follows from (18). Substituting (21) into (20) gives

$$\begin{aligned}
\nabla_{x^{(k)}} Q_{\text{noisy}}^{\pi_\theta}(s, x^{(k)}, k) &= \mathbb{E}_{a \sim \pi_\theta(\cdot \mid s, x^{(k)}, k)} \left[ Q^{\pi_\theta}(s, a)\, \nabla_{x^{(k)}} \log q_k(x^{(k)} \mid a) \right] - Q_{\text{noisy}}^{\pi_\theta}(s, x^{(k)}, k)\, \mathbb{E}_{a \sim \pi_\theta(\cdot \mid s, x^{(k)}, k)} \left[ \nabla_{x^{(k)}} \log q_k(x^{(k)} \mid a) \right] \\
&= \mathbb{E}_{a \sim \pi_\theta(\cdot \mid s, x^{(k)}, k)} \left[ \left( Q^{\pi_\theta}(s, a) - Q_{\text{noisy}}^{\pi_\theta}(s, x^{(k)}, k) \right) \nabla_{x^{(k)}} \log q_k(x^{(k)} \mid a) \right],
\end{aligned}$$

which is exactly the claim of Theorem 3.4.

# B. Experimental Details

## B.1. Experimental Setup

We implement NSPG using `JAX` with the `Flax` neural network library and `Optax` for optimization, enabling efficient just-in-time compilation and large-scale vectorized computation. All experiments are conducted on a machine equipped with **four NVIDIA L40 GPUs**.

For each environment, we train the agent for a fixed budget of **2 million gradient steps** to ensure convergence. We run **four independent random seeds** for every experimental configuration. Training is performed without early stopping, and all reported results correspond to fully trained policies rather than intermediate checkpoints. At test time, given a state observation, the diffusion actor generates multiple candidate actions by decoding from independent noise samples. The final action is selected based on the critic's action-value estimates.

The overall implementation follows an actor–critic structure, where diffusion-specific components are integrated into the policy optimization loop while maintaining a strict separation between noisy latent-space optimization and action-space value estimation.

## B.2. Evaluation Protocol

We evaluate policies periodically during training at fixed intervals of **10,000 gradient steps**. At each evaluation point, the current policy is executed in the environment over 10 episodes, and the average episodic return is recorded.

For each experiment, we aggregate evaluation results over the **final 50,000 training steps** across all random seeds and report the mean performance as the final score. This evaluation protocol avoids early-stopping selection and reflects the true performance of converged policies. All reported results are averaged across seeds and expressed using the same benchmark scores to facilitate direct comparison with prior works.

## B.3. NSPG Agent Architecture

NSPG follows an actor–critic design with diffusion policies, maintaining a strict separation between (i) *latent-space* policy optimization and (ii) *action-space* value estimation.

**Diffusion actor.** The actor is a conditional denoising diffusion model parameterized as a DDPM. For each state $s$, the model predicts the forward-process noise $\epsilon_\theta(s, x_t, t)$ given a noisy action latent $x_t$ and diffusion timestep $t \in \{1, \ldots, T\}$. Time conditioning is implemented using fixed Fourier features followed by a lightweight MLP time processor. The noise predictor is an MLP with Mish activations and hidden widths $(256, 256, 256, 256)$. Unless otherwise stated, we use $T = 5$ diffusion steps and a variance-preserving (VP) noise schedule. During training, noisy latents are constructed using the forward process

$$x_t = \sqrt{\bar{\alpha}_t}\, a + \sqrt{1 - \bar{\alpha}_t}\, \epsilon, \qquad \epsilon \sim \mathcal{N}(0, I),$$

where $a$ is the dataset action and $\bar{\alpha}_t$ is the cumulative product of diffusion alphas.

**Action-space critic ensemble.** The critic is an ensemble of $H$ Q-functions $Q_{\phi_h}(s, a)$ operating strictly on executed actions $a \in \mathcal{A}$. Unless otherwise stated, we use $H = 10$ critics with the same MLP architecture as the actor backbone (Mish activations, hidden widths $(256, 256, 256, 256)$). For targets and evaluation-time selection, we aggregate values by first averaging across the ensemble and then applying the configured target aggregation (e.g., LCB).

**Target networks and stabilization.** We maintain target networks for both the actor and critic via exponential moving average updates with rate $\tau_{\text{EMA}} = 0.005$. The critic target is updated after every gradient step. The actor target is updated every few steps after an initial warm-up period, following the same scheduling used throughout all experiments. This setup improves stability without changing the underlying objectives.

**Action sampling for training and evaluation.** To produce executed actions, the actor decodes using a reverse diffusion sampler (DDPM by default; DDIM is supported). At evaluation time, we sample $N_a = 10$ candidate actions per state and select the final action using the critic scores. Importantly, the critic is never evaluated on noisy latents; all value estimation is performed in action space.

## B.4. Noisy-Space Policy Gradient Estimation

NSPG optimizes diffusion policies by estimating gradients of a noisy-space action-value function, which assigns value to a noisy latent $x^{(k)}$ through the distribution of executed actions induced by the reverse diffusion process.

For a given state $s$, noisy latent $x^{(k)}$, and diffusion step $k$, we calculate the noisy-space action-value function $Q_{\text{noisy}}(s, x^{(k)}, k)$ using Monte Carlo sampling. Specifically, we draw $N$ executed actions $\{a_i\}_{i=1}^N$ by running the reverse diffusion process starting from $(x^{(k)}, k)$ down to step zero. Each action $a_i$ is evaluated using the target critic, and the noisy-space value is estimated as the sample mean of the resulting action-space Q-values.

To compute the noisy-space policy gradient, we employ a score-function estimator derived from the forward diffusion kernel. For each sampled action, we compute the score

$$\nabla_{x^{(k)}} \log q_k\left(x^{(k)} \mid a\right) = -\frac{x^{(k)} - \sqrt{\bar{\alpha}_k}\, a}{1 - \bar{\alpha}_k},$$

and weight it by the corresponding noisy-space advantage

$$A_{\text{noisy}}(s, a, x^{(k)}, k) = Q(s, a) - Q_{\text{noisy}}(s, x^{(k)}, k).$$

The final gradient is obtained by averaging the score-weighted advantages across Monte Carlo samples.

This procedure yields an estimate of $\nabla_{x^{(k)}} Q_{\text{noisy}}(s, x^{(k)}, k)$ without differentiating through the reverse diffusion trajectory and without evaluating the critic on noisy latent variables, preserving both computational efficiency and semantic correctness.

**Actor and Critic Optimization**    The actor is optimized using a combination of a denoising objective and a noisy-space value guidance term. The denoising objective corresponds to standard diffusion training and encourages the actor to predict the forward-process noise from dataset actions. This term serves as a behavior cloning regularizer that anchors the policy to the support of the offline data.

Policy improvement is incorporated through a *soft noisy-space guidance* term, which injects the noisy-space policy gradient into the diffusion regression objective. Concretely, the guidance signal is given by the inner product between the predicted noise and the estimated noisy-space gradient, scaled by the diffusion noise level $\sqrt{1 - \bar{\alpha}_k}$. The resulting actor objective balances imitation and value-driven improvement while remaining compatible with the diffusion training formulation. In all experiments, we exclusively use soft guidance.

The critic is trained using Bellman regression on offline data. Target values are computed using actions sampled from the target actor and evaluated by a target critic ensemble. To mitigate overestimation and distributional shift, we employ a pessimistic target construction based on a lower-confidence bound (LCB), which subtracts a multiple of the ensemble standard deviation from the mean Q-value. Both actor and critic are optimized using Adam-based optimizers with gradient clipping set to 1. Target networks are maintained via exponential moving average updates to improve training stability.

*Table 3.* Shared hyperparameters used across all tasks and experiments.

| Hyperparameter | Value |
| --- | :---: |
| Diffusion steps $T$ | 5 |
| Noise schedule $\beta_k$ | Variance preserving (VP) |
| Critic ensemble size $H$ | 10 |
| Batch size | 256 |
| Actor learning rate | $3 \times 10^{-4}$ |
| Critic learning rate | $3 \times 10^{-4}$ |
| Learning rate schedule | None (constant) |
| Optimizer | AdamW |
| $N$ (NSPG samples) | 4 |
| Discount factor $\gamma$ | 0.99 |
| EMA rate $\tau_{\text{EMA}}$ | 0.005 |
| Number of sampled actions (evaluation) $N_a$ | 10 |
| Number of Q samples (target computation) | 10 |
| Action clipping | Enabled |

## B.5. Hyperparameters

We use a largely shared set of hyperparameters across all tasks and experimental settings to ensure consistency and fair comparison. These fixed hyperparameters cover the diffusion configuration, network architectures, optimization settings, and evaluation parameters, and are kept identical across experiments unless explicitly stated otherwise. The shared hyperparameters are summarized in Table 3.

In addition to the shared configuration, we vary a small number of hyperparameters across experiments to study their impact on performance and stability. These experiment-specific hyperparameters primarily control the strength of noisy-space policy improvement, behavior cloning regularization, and pessimism in value estimation. We summarize these settings in Table 4, where each row corresponds to a specific experiment and only the modified hyperparameters are listed.

*Table 4.* Experiment-specific hyperparameters varied across tasks. OGBench task names are shortened for readability.

| Domain | Task | $\lambda$ | $\eta$ | $\rho$ |
|---|---|---|---|---|
| Locomotion | HalfCheetah-medium-v2 | 50 | 1 | 0 |
| | Hopper-medium-v2 | 30 | 1 | 1 |
| | Walker2d-medium-v2 | 3 | 1 | 1 |
| | HalfCheetah-medium-replay-v2 | 100 | 1 | 0 |
| | Hopper-medium-replay-v2 | 30 | 1 | 1 |
| | Walker2d-medium-replay-v2 | 10 | 1 | 1 |
| | HalfCheetah-medium-expert-v2 | 10 | 1 | 0 |
| | Hopper-medium-expert-v2 | 1 | 1 | 0 |
| | Walker2d-medium-expert-v2 | 30 | 1 | 1 |
| AntMaze | AntMaze-medium-play-v0 | 1 | 0.1 | 1.3 |
| | AntMaze-medium-diverse-v0 | 5 | 0.1 | 1.3 |
| | AntMaze-large-play-v0 | 5 | 0.1 | 1 |
| | AntMaze-large-diverse-v0 | 1 | 0.1 | 0.8 |
| Adroit | Pen-Human-v1 | 1 | 3 | 0 |
| | Pen-Cloned-v1 | 1 | 10 | 0 |
| OGBench Visual | visual-cube-single | 0.3 | 0.3 | 0 |
| | visual-cube-double | 0.3 | 0.3 | 0 |
| | visual-puzzle-3x3 | 5 | 0.3 | 0 |
| | visual-puzzle-4x4 | 3 | 0.3 | 0 |

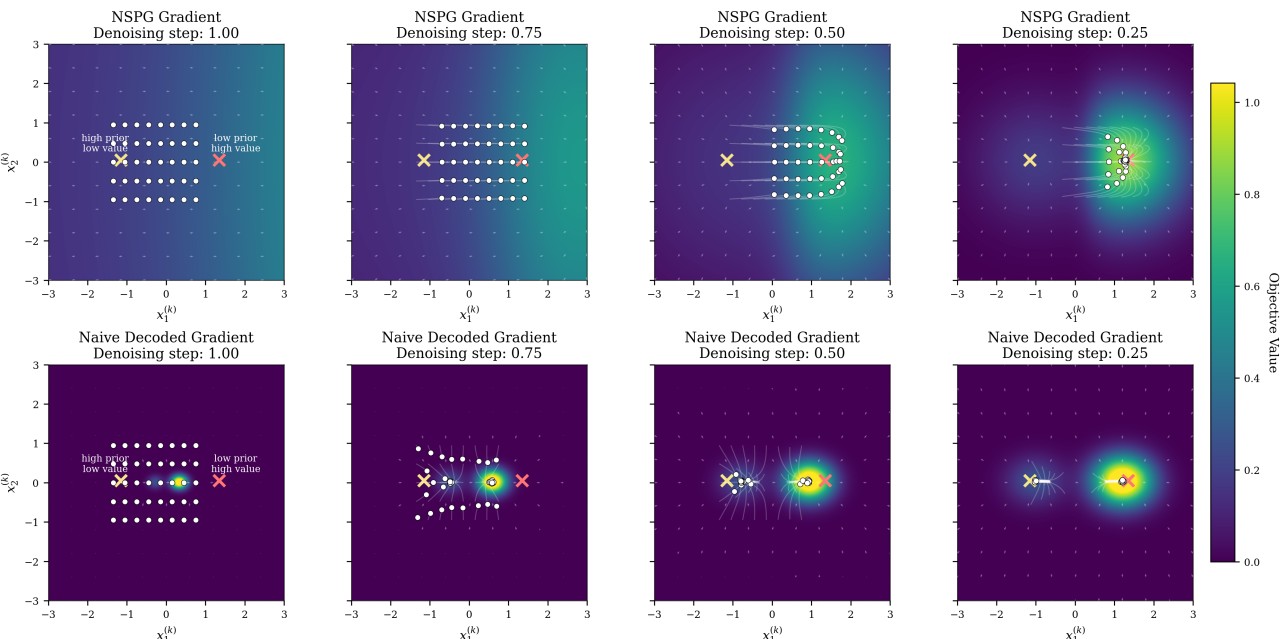

*Figure 4.* **Toy illustration of noisy-space gradients across denoising steps.** We consider a two-dimensional multimodal action distribution with a high-density low-reward region and a lower-density high-reward region. Denoising proceeds from the noisiest step, shown as 1, toward the clean sample, corresponding to step 0. Top: NSPG follows $\nabla_{x^{(k)}} Q_{\text{noisy}}(x^{(k)}, k)$, the gradient of the expected clean-action value under the diffusion-induced conditional. Bottom: the decoded baseline follows $\nabla_{x^{(k)}} Q(\hat{a}(x^{(k)}))$ after mapping each noisy latent to a single clean action. Columns show snapshots at different denoising steps; arrows denote gradient fields and traces show illustrative gradient-ascent rollouts.

## B.6. Toy Example: Noisy-Space Value Gradients

We illustrate the semantics of noisy-space policy gradients in a simple two-dimensional example. The goal is to visualize how different ways of assigning clean-action values to noisy latents induce different gradient fields. In particular, we compare the proposed NSPG field, which is defined through the diffusion-induced surrogate distribution over clean actions, to a decoded gradient that first maps each noisy latent to a single clean action.

**Setup.** The clean action space is $a \in \mathbb{R}^2$. We define a bimodal reference action distribution over clean actions,

$$\mu(a) = 0.8\,\mathcal{N}(a; \mu_0, \Sigma_0) + 0.2\,\mathcal{N}(a; \mu_1, \Sigma_1), \tag{22}$$

where $\mu_0, \mu_1 \in \mathbb{R}^2$ and $\Sigma_0, \Sigma_1 \in \mathbb{R}^{2 \times 2}$ are fixed means and covariances. Thus, the first component represents a high-density action region, while the second component has lower probability mass.

We define a clean-action value landscape using the same Gaussian components but with different coefficients,

$$Q(a) = 0.55\,\mathcal{N}(a; \mu_0, \Sigma_0) + 2.35\,\mathcal{N}(a; \mu_1, \Sigma_1), \tag{23}$$

In the visualization, the second mode is assigned a substantially larger value coefficient than the first. This creates a controlled multimodal setting in which the higher-density action region has lower value, while the lower-density region has higher value. The magnitude of this value difference is intentionally chosen to make the effect of value guidance visible; changing the relative coefficients would change the balance between preserving prior mass and moving toward the higher-value region.

**Surrogate semantics in noisy space.** For a diffusion step $k$, clean actions are mapped to noisy latents through the forward diffusion kernel

$$q_k(x^{(k)} \mid a) = \mathcal{N}\left(x^{(k)}; \sqrt{\bar{\alpha}_k}\,a, (1 - \bar{\alpha}_k)I\right), \qquad 0 < \bar{\alpha}_k < 1. \tag{24}$$

Following the surrogate semantics used in the main text, we define the joint distribution

$$\tilde{p}_k(a, x^{(k)}) = \mu(a)\, q_k(x^{(k)} \mid a), \tag{25}$$

which induces the conditional distribution

$$\tilde{p}_k(a \mid x^{(k)}) = \frac{\mu(a) q_k(x^{(k)} \mid a)}{\int \mu(a') q_k(x^{(k)} \mid a')\, da'}. \tag{26}$$

This conditional captures the clean actions compatible with a noisy latent under the forward diffusion process. Thus, a noisy latent is not assigned to a single clean action; it is associated with a distribution over clean actions that could have generated it.

**NSPG objective.** The noisy-space value of a latent is the expected clean-action value under this diffusion-induced conditional:

$$Q_{\text{noisy}}(x^{(k)}, k) = \mathbb{E}_{a \sim \tilde{p}_k(\cdot \mid x^{(k)})}\left[Q(a)\right]. \tag{27}$$

The NSPG field visualized in the figure is

$$\nabla_{x^{(k)}} Q_{\text{noisy}}(x^{(k)}, k). \tag{28}$$

In the Gaussian-mixture toy setting, the conditional distribution in (26) and the expectation in (27) can be evaluated analytically. We therefore compute $Q_{\text{noisy}}$ exactly on a grid and differentiate the resulting scalar field numerically for visualization. This avoids Monte Carlo noise in the plotted fields. In the full NSAC algorithm, by contrast, $Q_{\text{noisy}}$ and its gradient are estimated using Monte Carlo samples from the diffusion policy.

**Decoded gradient baseline.** For comparison, we visualize a decoded gradient field. This baseline maps each noisy latent to a single clean action by inverting the mean of the forward process,

$$\hat{a}(x^{(k)}) = \frac{x^{(k)}}{\sqrt{\bar{\alpha}_k}}, \tag{29}$$

and then differentiates the clean-action value at the decoded action:

$$\nabla_{x^{(k)}} Q(\hat{a}(x^{(k)})). \tag{30}$$

This construction is well-defined, but it depends on a deterministic choice of decoder. It therefore optimizes the value of a single decoded action rather than the expected clean-action value under the diffusion-induced conditional in (26).

**Visualization across denoising steps.** Figure 4 shows snapshots of the two gradient fields across several denoising steps. The top row visualizes the NSPG field and the bottom row visualizes the decoded gradient field. Columns correspond to decreasing denoising steps, moving from noisier latents toward cleaner latents. The background shows the corresponding scalar objective value, the arrows show the gradient field, and the traces show illustrative gradient-ascent rollouts under the plotted fields.

The figure is intended as an illustrative diagnostic rather than a claim about global convergence. In this particular toy configuration, the higher-value mode is assigned a sufficiently large coefficient, so the NSPG field induces a strong bias toward the high-value region for the shown initialization. This does not imply that NSPG always collapses to a single mode. Rather, the behavior depends on the relative value coefficients, the prior density of each mode, the diffusion noise level, and the current latent location. The purpose of the example is to show that NSPG and decoded gradients can produce different noisy-space geometries because they optimize different scalar objectives.

**Takeaway.** This example highlights that lifting value gradients from clean action space to noisy latent space is not unique. A decoded gradient uses a deterministic assignment from noisy latents to clean actions, while NSPG uses the surrogate distribution induced by the forward diffusion process. Consequently, NSPG optimizes an expected clean-action value over compatible actions, whereas the decoded gradient optimizes the value of a single decoded action. The resulting gradient fields can differ substantially, especially in multimodal settings where probability mass and value are misaligned.

# C. Additional Experimental Results

This appendix provides additional experimental results that complement the main paper. Specifically, we report (i) full training curves across all environments, (ii) ablation studies on the pessimism coefficient $\rho$ and critic ensemble size, and (iii) an analysis of training time and computational overhead. Unless stated otherwise, all results follow the experimental setup described in Section 6.

## C.1. Training Dynamics

Figure 5 shows learning curves for NSAC across all 15 evaluation environments, including locomotion, AntMaze, and Adroit tasks. Curves report average normalized return over environment steps, with shaded regions indicating standard deviation across seeds.

Across domains, NSAC exhibits stable training dynamics and consistent convergence behavior. In more challenging environments, such as AntMaze and dexterous manipulation tasks, learning progress is slower but remains stable, reflecting the increased difficulty of long-horizon and sparse-reward settings.

## C.2. Effect of Pessimism Coefficient $\rho$

We study the sensitivity of NSAC to the pessimism coefficient $\rho$, which controls the degree of conservatism in the lower-confidence bound critic. Figure 6 reports performance across representative environments for different values of $\rho$.

Across tasks, NSAC demonstrates robust performance over a broad range of pessimism levels. Moderate values of $\rho$ provide the best balance between avoiding overestimation and preserving learning signal, while excessively small or large values can degrade performance. These results indicate that NSAC does not require precise tuning of the pessimism coefficient to achieve strong performance.

## C.3. Effect of Critic Ensemble Size

We further analyze the effect of the critic ensemble size on performance. Figure 7 shows results for different ensemble sizes across two representative environments.

Performance generally improves with small ensemble sizes, reflecting more reliable value estimation, but saturates beyond a modest number of critics. This suggests that NSAC benefits from ensemble-based pessimism without requiring large ensembles, keeping computational overhead manageable.

## C.4. Training Time and Computational Cost.

*Table 5.* Computational cost of NSAC for different numbers of Monte Carlo samples, compared to DAC. Wall-clock time is normalized by DAC.

| Method | Time / Update (ms) | Wall-clock Relative to DAC |
|---|---|---|
| NSAC ($n_{\text{samples}} = 2$) | 3.2 | 0.90$\times$ |
| NSAC ($n_{\text{samples}} = 4$) | 3.4 | 0.98$\times$ |
| NSAC ($n_{\text{samples}} = 8$) | 3.9 | 1.07$\times$ |
| NSAC ($n_{\text{samples}} = 16$) | 5.8 | 1.40$\times$ |
| DAC (Fang et al., 2024) | 3.6 | 1.00$\times$ |

We compare the computational cost of NSAC with DAC (Fang et al., 2024) under a matched implementation using the same network architecture, diffusion horizon, and hardware setup. As shown in Table 5, NSAC with two Monte Carlo samples is faster than DAC, requiring 3.2 ms per update and $0.9\times$ the wall-clock time of DAC. With four samples, NSAC remains comparable to DAC, requiring 3.4 ms per update and $0.98\times$ relative wall-clock time. Increasing the number of samples leads to higher cost, but provides limited empirical benefit, as discussed in the Monte Carlo ablation. These results indicate that the noisy-space expectation can be estimated efficiently in practice, and that NSAC does not introduce prohibitive computational overhead compared to existing diffusion actor–critic methods.

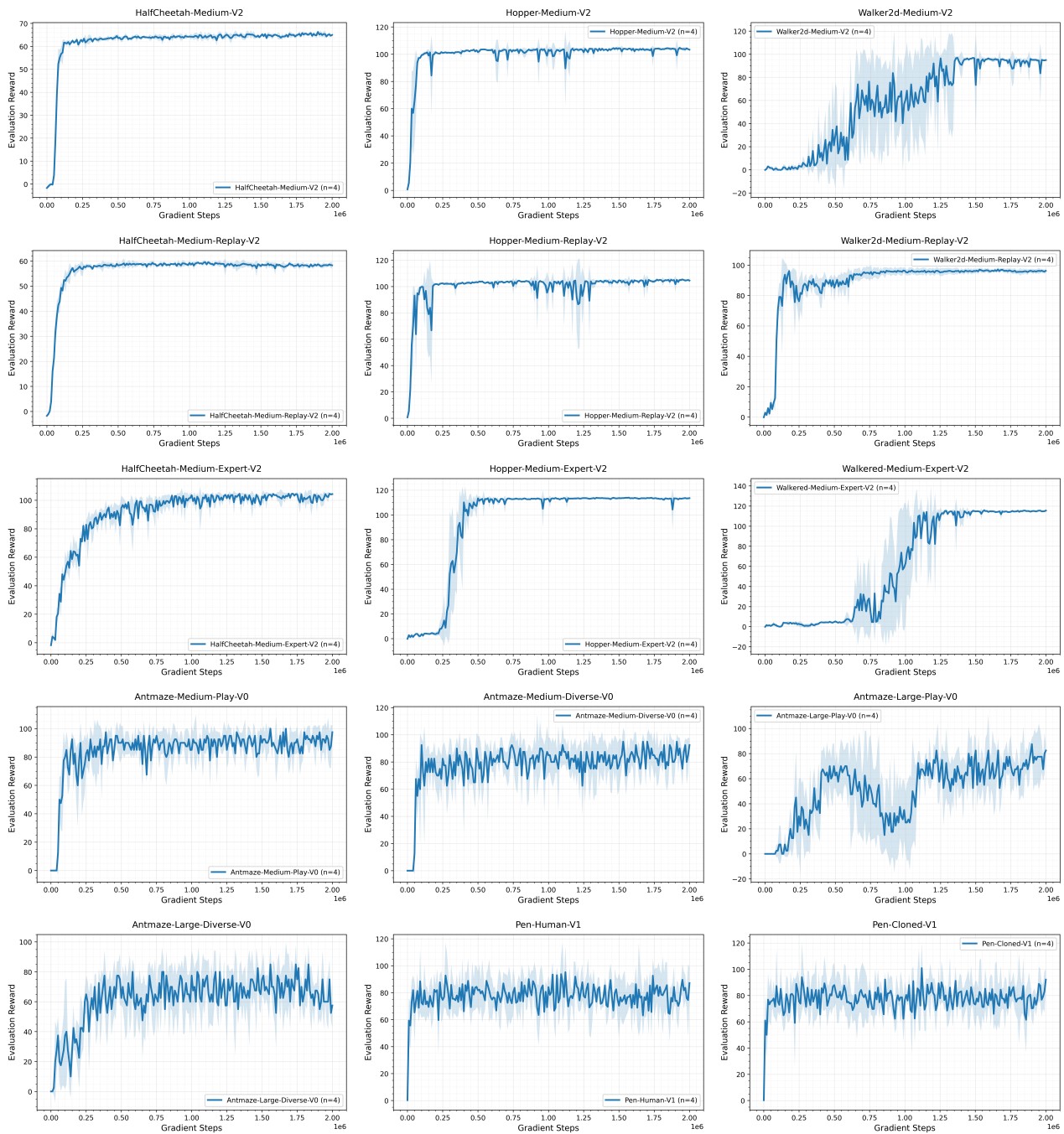

*Figure 5.* Training curves for NSAC across all 15 evaluation environments. Each subplot corresponds to one environment and reports normalized return as a function of training steps. Shaded regions indicate standard deviation across seeds.

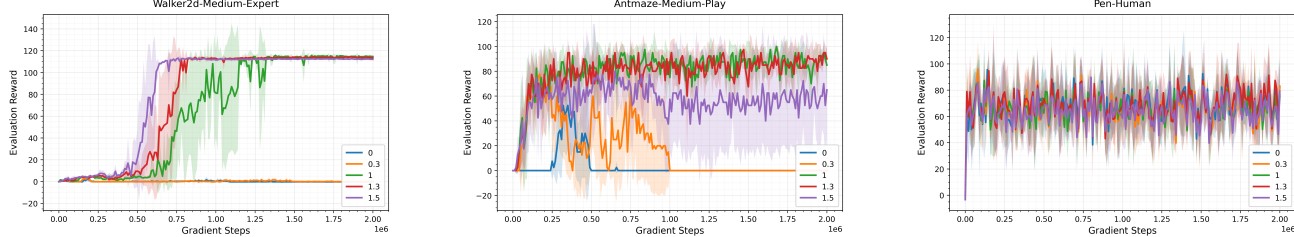

*Figure 6.* Ablation on the pessimism coefficient $\rho$ across three representative environments. Curves report normalized return as a function of training steps for different values of $\rho$.

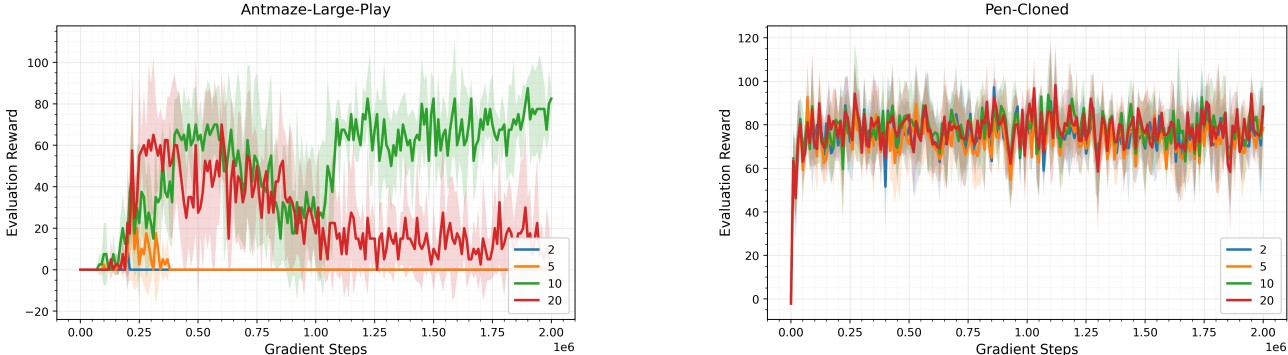

*Figure 7.* Ablation on critic ensemble size across two representative environments. Curves report normalized return for different ensemble sizes.

