# OpenReview forum: "Noisy-Space Policy Gradient for Diffusion Policies in Offline Reinforcement Learning"
_ICML.cc/2026/Conference — ICML 2026 regular_

### Official Review · Reviewer_bj8W · 2026-02-23

**Soundness:** 2
**Presentation:** 3
**Significance:** 3
**Originality:** 3
**Overall Recommendation:** 4
**Confidence:** 3

**Summary:**

The paper introduces a noisy-space action-value function intended to be semantically aligned with the interface between diffusion policies and the environment, and derives Noisy-Space Policy Gradient (NSPG) that avoids backpropagation through the full denoising trajectory. Building on this formulation, the authors further propose a Noisy-Space Actor–Critic (NSAC) algorithm for offline reinforcement learning, which leverages gradients of the noisy-space action-value function for diffusion-policy improvement while learning the critic in the original action space. Experiments on the D4RL benchmark validate the effectiveness of the proposed method.

**Compliance With Llm Reviewing Policy:**

Affirmed.

**Final Justification:**

My concerns have been adequately addressed. I will retain my positive feedback. A major concern is the practicality of the proposed method, including its convergence upper bound and the feasibility of its online variant.

**Key Questions For Authors:**

- Since the NSPG framework is only validated in the offline setting, can the authors provide its feasibility or the main bottlenecks in an online implementation?
- Can you explain Definition 3.3 more clearly? It is a littile confusing.

**Limitations:**

See Weaknesses and Questions above.

**Strengths And Weaknesses:**

Strengths:
- The paper is well written, and the formal statements are clear.
- It addresses a significant issue of the mismatch between the value function expected to evaluate noisy latents and the reward explicitly evaluating clean actions.
- The proposed method is original, with a detailed exposition of its relationship to prior work.

Weaknesses:
- The results on the locomotion tasks in Table 1 are not significant compared with those of DAC. It would be helpful to include additional quantitative metrics (e.g. training stability) for a more comprehensive analysis. In addition, the ablation studies are somewhat limited. For instance, since the number of Monte Carlo samples plays a key role in estimating $Q_{noisy}$, it would be useful to characterize the trade-off between convergence performance and training overhead as this parameter varies.
- The paper does not clearly justify why avoiding backpropagation through the denoising process is important in the offline setting.

---

> ### Author Rebuttal · Authors · 2026-03-31
>
> We thank the reviewer for their thoughtful feedback and appreciate their recognition of our clarity, originality, and positioning relative to prior work, as well as the importance of addressing the mismatch between noisy and action-space value functions.
>
> Below, we address the raised concerns:
>
> ---
>
> ## Clarifications on The Weaknesses part.
>
> **(A) Empirical significance on some benchmarks and evaluation scope**
>
> We thank the reviewer for raising this point. While gains on locomotion tasks may appear modest, this reflects the broader context: D4RL locomotion benchmarks are relatively saturated, with typically incremental improvements. In contrast, we observe consistent and substantial gains on more challenging settings—particularly **AntMaze and Adroit manipulation tasks**, which emphasize long-horizon credit assignment and multimodality—where our method significantly outperforms DAC, with improvements exceeding **50% on large AntMaze and 30% on pen-cloned tasks**.
>
> ---
>
>
> **(B) Ablations, stability, and Monte Carlo tradeoff**
>
> We thank the reviewer for highlighting the importance of ablations and the trade-off between convergence and computational overhead. The paper includes ablations on key design choices: guidance strength (`Fig. 1`), number of Monte Carlo samples (`Fig. 2`), conservatism strength (`Fig. 5`), and critic ensemble size (`Fig. 4`). We additionally provide ablations on behavior cloning (BC) regularization strength below:
>
> |BC strength (η) |Reward (cheetah-m-r)|
> |-|-|
> |0.1|59.0|
> |0.3|58.3|
> |1|59.1|
> |3|59.4|
> |10|59.2|
>
>
> To clarify the trade-off between performance, stability, and computational cost, we provide additional analysis on the pen-cloned environment varying the number of Monte Carlo samples. We report final return, cross-seed standard deviation, and standard deviation over the last 5 checkpoints (as a proxy for training stability), along with time per update, relative wall-clock time, and peak GPU memory to capture computational overhead:
>
> |Method|Final return|Std across seeds|Time/Update (ms)|Wall-clock (relative to DAC)|Peak GPU Memory (GB)|Std across last 5 checkpoints|
> |-|-|-|-|-|-|-|
> |NSPG(2 Samples)|82.9|3.0|3.2|0.9x|2.64|8.5|
> |NSPG(4 Samples)|84.4|3.1|3.4|0.98x|2.64|6.3|
> |NSPG(8 Samples)|84.6|1.93|3.9|1.07x|2.65|7.8|
> |NSPG(16 Samples)|84.5|3.3|5.8|1.4x|2.65|7.5|
> |DAC|63.9|7.3|3.6|1x|2.8|8.4|
>
> Across multiple environments, 2–4 samples are sufficient for strong performance, with diminishing returns beyond this range. This regime also maintains stable training, as supported by the learning curves (`Fig. 4`) and the std metrics.
>
> ---
>
> **(C) Role of avoiding backpropagation through denoising**
>
> We thank the reviewer for raising this important point. Avoiding backpropagation through the full denoising process is a key design choice for both computational and optimization reasons. In diffusion policies, this would require differentiating through a sequence of stochastic transformations—akin to backpropagation through time—resulting in high-variance gradients, increased memory usage, and slower, less stable training.
>
> Our approach instead leverages a noisy-space value function to perform a single-step regression toward a target score. This leads to a more stable and efficient optimization and avoids querying the critic on intermediate noisy actions that may lie outside the data distribution, a key issue in offline RL.
>
> We will clarify this further distinction in the final.
>
> ## Response To The Questions
>
>
> **(1) Online setting**
> We thank the reviewer for this question. The framework can be naturally extended to online RL: newly collected experience can be added to the replay buffer, and the policy and critic updated as in standard off-policy methods. This aligns with prior work such as FQL, which extends flow policies to online settings via a similar mechanism. We leave this to future work and will clarify this in the final version.
>
> ---
>
> **Clarification of Definition 3.3.** We thank the reviewer for pointing out that this definition could be clearer. Intuitively, Definition 3.3 introduces a surrogate joint distribution over noisy and clean actions $(a, x^{(k)})$, where a clean action is first sampled from the policydistribution (the diffusion model) and then mapped to a noisy latent via the forward diffusion process.
>
> Importantly, diffusion policies define a generative denoising process from noise to actions but do not explicitly specify a joint distribution over intermediate noisy latents and executable actions. The surrogate joint resolves this by linking noisy latents to executable actions via the forward diffusion kernel, ensuring value estimation remains in the clean action space where rewards are defined, rather than evaluating potentially out-of-distribution noisy samples
>
> ---
>
> We thank the reviewer for the thoughtful feedback. We hope our responses have addressed the main concerns. We are happy to provide any further clarification to any further questions the reviewer may have.

---

> > ### Author Rebuttal · Reviewer_bj8W · 2026-04-02
> >
> > My concerns have been adequately addressed. I will retain my positive feedback.

---

### Official Review · Reviewer_5A1x · 2026-03-10

**Soundness:** 3
**Presentation:** 4
**Significance:** 3
**Originality:** 4
**Overall Recommendation:** 4
**Confidence:** 3

**Summary:**

This paper introduces Noisy-Space Actor-Critic (NSAC), a novel approach for training diffusion policies in offline reinforcement learning. The authors identify a fundamental mismatch in existing diffusion RL methods: policy optimization occurs in a noisy latent space, while Q-values are defined exclusively on clean, executed actions. To bridge this gap, they define a "noisy-space action-value function" that represents the expected action-space return conditioned on a noisy latent variable. By introducing a surrogate joint distribution linking clean actions to noisy latents via the forward diffusion kernel, the authors derive a Noisy-Space Policy Gradient (NSPG). This allows for value-guided policy improvement at each denoising step without backpropagating through the entire denoising chain and without erroneously querying the critic on noisy, out-of-distribution actions. The method is evaluated on the D4RL benchmark.

**Compliance With Llm Reviewing Policy:**

Affirmed.

**Final Justification:**

My final recommendation remains weak accept. I continue to view the paper as a technically strong, original, and clearly presented contribution, and the rebuttal further strengthened my confidence in its value. In particular, the authors provided helpful clarifications and additional empirical support, which reinforced my overall assessment rather than changing it substantially. Overall, I remain the positive feedback.

**Key Questions For Authors:**

1. What is the exact computational cost (wall-clock time and memory) of training NSAC compared to methods that just re-weight the final clean actions? Specifically, how many Monte Carlo samples are required to keep the variance of the NSPG estimator manageable?
2. The Surrogate Noisy Joint relies on the forward diffusion kernel. As the policy is updated to maximize Q-values, the reverse process deviates from the true inverse of the forward process. How does this structural shift affect the theoretical validity of your noisy-space gradient late in training?
3. Since D4RL is a highly saturated benchmark, can you provide results on more complex, high-dimensional tasks (e.g., vision-based control or highly multimodal environments) to prove that the expressive power of NSAC genuinely exceeds simpler action-space guidance methods?

**Limitations:**

The theoretical assumptions regarding the surrogate semantic model are clearly stated. However, the authors fail to sufficiently address the severe computational cost and high variance associated with Monte Carlo estimation of the noisy-space advantage during training. The limitations section must explicitly discuss the trade-off between the theoretical purity of the NSPG and the practical wall-clock time required to compute it.

**Strengths And Weaknesses:**

Strengths:
1. The paper identifies a highly relevant and often-ignored theoretical flaw in diffusion-based RL: the semantic mismatch between noisy latent states and action-space value functions.
2. The mathematical formulation is elegant. Deriving a noisy-space policy gradient that depends only on clean-action Q-values bypasses the catastrophic extrapolation errors that occur when critics are forced to evaluate intermediate noisy actions.
3. Avoiding backpropagation through the full denoising chain makes the method significantly more computationally stable than end-to-end differentiable diffusion RL methods.

Weaknesses:
1. Computational Burden of Monte Carlo Estimation: The gradient requires computing the expected noisy-space advantage, which relies on Monte Carlo sampling of clean actions from the current noisy latent. Running multiple full reverse-diffusion passes for every gradient step to accurately estimate this expectation seems computationally prohibitive. If the number of action samples is small, the variance of this gradient estimate could be severe.
2. Validity of the Surrogate Semantics: The theoretical derivations rely heavily on the "Surrogate Noisy Joint", which assumes the distribution of clean actions given a noisy latent is perfectly characterized by the forward diffusion kernel. However, under value-guided optimization, the reverse process distribution inevitably shifts away from the exact forward marginals. The paper does not adequately analyze how this distribution shift degrades the accuracy of the NSPG.
3. Saturated Baselines: The empirical claims rest entirely on the D4RL benchmark, which is heavily saturated. The paper claims consistent performance gains, but given the heavy mathematical machinery, marginal improvements on D4RL are insufficient to prove the practical necessity of this method over simpler, action-space reweighting baselines.

---

> ### Author Rebuttal · Authors · 2026-03-31
>
> We thank the reviewer for the thoughtful and constructive feedback. We particularly appreciate the recognition of the core issue we address—the mismatch between noisy latent variables and action-space value functions in diffusion-based RL.
> We are also grateful for the positive assessment of our formulation, as well as the acknowledgment of the benefits of avoiding backpropagation through the full denoising process.
>
> We address the reviewer’s concerns below:
>
> ---
>
> ### (A) Computational Cost and Monte Carlo Estimation
>
> We thank the reviewer for raising this important concern. The manuscript includes an ablation over the number of MC samples (2, 4, 8, 16) across multiple environments (`Fig. 2`). These results show that 2–4 samples are sufficient to achieve near-saturated performance, with diminishing returns beyond that. Using two samples also yields stable learning dynamics, indicating that the variance of the estimator is well controlled in practice.
>
> To further clarify this trade-off, we provide additional analysis summarizing performance, training stability (via standard deviation across seeds and across checkpoints), and computational cost (total wall-clock time, time per gradient update, and peak GPU memory):
>
> |Method|Final return|Std across seeds|Time/Update (ms)|Wall-clock (relative to DAC)|Peak GPU Memory (GB)|Std across last 5 checkpoints|
> |-|-|-|-|-|-|-|
> |NSPG(2 Samples)|82.9|3.0|3.2|0.9x|2.64|8.5|
> |NSPG(4 Samples)|84.4|3.1|3.4|0.98x|2.64|6.3|
> |NSPG(8 Samples)|84.6|1.93|3.9|1.07x|2.65|7.8|
> |NSPG(16 Samples)|84.5|3.3|5.8|1.4x|2.65|7.5|
> |DAC|63.9|7.3|3.6|1x|2.8|8.4|
>
> These results show that NSPG achieves competitive or better performance with comparable (or lower) wall-clock cost, while maintaining stable training dynamics with only 2–4 samples. This indicates that Monte Carlo estimation does not introduce prohibitive overhead in practice.
>
> ---
>
> ### (B) Validity of the Surrogate Noisy-Space Semantics
>
> We thank the reviewer for this insightful point. The surrogate construction is not intended to assume that the learned reverse process exactly matches the forward-noising posterior. Rather, it provides a principled way to assign value to noisy latents through the clean actions they induce. Importantly, this allows NSPG to rely exclusively on clean action-space Q-values, avoiding the need to evaluate critics on noisy, out-of-distribution variables—a key challenge in offline RL.
>
> We note that, as the policy evolves, the reverse process may deviate from the forward marginals. In the offline RL setting however, this effect is mitigated—though not eliminated—by behavior regularization of the target policy, which constrain the learned policy toward the data distribution. Empirically, we observe stable training and consistent performance gains across environments (`Figs. 2, 4`) even with small MC samples, indicating that this approximation provides a reliable optimization signal in practice.
>
> We agree that a more detailed theoretical analysis of this distributional mismatch would be valuable, and we will clarify this discussion and highlight it as an important direction for future work.
>
> ### **(C) Empirical Scope and Benchmark Choice**
>
>
> We thank the reviewer for this comment. We agree that evaluating beyond D4RL is important, and note that our goal is to assess whether the proposed method provides consistent improvements across different settings.
>
> While D4RL remains a standard benchmark for offline RL, our experiments span diverse domains including locomotion, AntMaze, and high-dimensional manipulation (Adroit), where NSAC shows consistent gains. In particular, we observe more pronounced improvements on **AntMaze and Adroit manipulation tasks**, which involve long-horizon credit assignment and multimodal action distributions. In these regimes, NSAC shows clear advantages over prior methods, supporting our hypothesis that semantically grounded value guidance becomes increasingly important as task complexity grows.
>
> To further address this point, we provide additional results on visual control tasks from OGBench:
>
> | Environment | FQL | IQL | NSAC (ours) |
> |-|-|-|-|
> | visual-cube-single-play-singletask-task1-v0 | 81 | 70 | **89** |
> | visual-cube-double-play-singletask-task1-v0 | 21 | 34 | **36** |
>
> These environments involve higher-dimensional observations and more complex state representations. NSAC consistently outperforms prior methods, indicating that the benefits of noisy-space value guidance extend beyond standard D4RL benchmarks.
>
> We will include these results in the final version and further clarify the scope of evaluation.
>
> ---
>
> We thank the reviewer again for the thoughtful feedback. We hope our responses have adequately addressed the main concerns. We are happy to provide any further clarification or respond to any additional questions the reviewer may have.

---

> > ### Author Rebuttal · Reviewer_5A1x · 2026-04-03
> >
> > The rebuttal responds convincingly to the concern about computational cost and further strengthens the empirical support through additional analysis. That said, the main conceptual issue regarding the surrogate noisy-space interpretation is only partially addressed. The authors provide a plausible practical rationale, but the response still falls short of clarifying how the gap between the surrogate formulation and the learned reverse process influences the validity of the gradient at later stages of training. Overall, the rebuttal increases my confidence in the approach, but a key theoretical concern remains unresolved.

---

> > > ### Author Response · Authors · 2026-04-05
> > >
> > > We thank the reviewer for raising this important point and for the thoughtful follow-up.
> > >
> > > ---
> > >
> > > > The authors provide a plausible practical rationale, but the response still falls short of clarifying how the gap between the surrogate formulation and the learned reverse process influences the validity of the gradient at later stages of training.
> > >
> > > We thank the reviewer for highlighting this important point. Our clarification is that this gap does **not** make the gradient used by our method invalid at later stages of training. The surrogate semantics in Sec. 3.2 is introduced because a diffusion policy does not by itself specify the joint structure between clean actions and intermediate noisy latents needed to define this gradient; the surrogate provides that missing semantic bridge, but is not meant to assert that the learned reverse process exactly matches the forward posterior.
> > >
> > > More precisely, the surrogate is a deliberately constructed semantic model that couples a reference action distribution with the known forward diffusion kernel, and this joint exists by construction. In general, diffusion training guarantees the final action marginal, but not the clean/noisy joint needed for Bayes conditioning on intermediate latents; our surrogate supplies that missing structure using the forward kernel. This is also consistent with diffusion modeling itself, which already introduces a forward corruption process as a modeling device rather than inferring such structure from data.
> > >
> > > The relevant issue is therefore not existence, but whether this surrogate joint coincides with the joint induced by the learned reverse process. In the offline RL setting, any resulting mismatch is kept controlled by the denoising / behavior-cloning term and policy regularization, which anchor the learned policy to the data-supported action distribution and limit uncontrolled drift away from the surrogate conditioning semantics. Empirically, if this mismatch became harmful late in training, one would expect  late-stage degradation; instead, `Fig. 4` shows no systematic late-stage deterioration across environments. We will revise the manuscript to make this distinction clearer.
> > >
> > > We thank the reviewer again for raising this important point, and we hope this clarification better addresses the concern regarding the validity of the gradients at later stages of training.

---

### Official Review · Reviewer_oMgq · 2026-03-12

**Soundness:** 3
**Presentation:** 3
**Significance:** 2
**Originality:** 2
**Overall Recommendation:** 4
**Confidence:** 4

**Summary:**

This paper addresses the mismatch issue in flow-based reinforcement learning, i.e., while policies are typically modeled in the noise space and generate actions via denoising, value functions are defined in the clean action space. Such a mismatch may lead to a lack of clear guidance signals for intermediate flow steps, resulting in difficult learning. To this end, the paper proposes a noise-space Q-function that maps noisy intermediate variables to Q-values. Based on this, the paper derives a noise-space policy gradient method in the framework of the flow-based reinforcement learning.

**Compliance With Llm Reviewing Policy:**

Affirmed.

**Final Justification:**

The author has provided detailed responses to each of my concerns and analyzed that the essential difference between existing methods. I agree with this explanation and therefore intend to raise my score to 4.

**Key Questions For Authors:**

It appears to me that the proposed method is essentially a variant of classifier-free guidance, where the conventional value-based guidance is reformulated in the noisy latent space.

The main novelty seems to lie in absorbing both the generative score and the guidance signal into a single score network during training, rather than combining them explicitly at inference time.

I would appreciate clarification on whether this modification leads to any fundamental theoretical or optimization advantage, or whether it mainly represents a different parameterization of the same guidance principle.

**Limitations:**

yes

**Strengths And Weaknesses:**

Pros:

1.	The method is theoretically motivated and optimizes the training procedure. By introducing Q-values in the noise space, it provides stronger guidance signals for intermediate flow steps, thus improving the stability of diffusion models.

2.	Experiments verify the effectiveness of the proposed method. Figures 1 and 2 show the impact of different settings on performance, and Figure 3 provides examples that illustrate the differences between noise-space gradients and traditional gradients.

Cons:

1.	The motivation is not clearly articulated. Existing diffusion or flow-based methods usually introduce reward guidance after action generation by weighting the action distribution with reward signals, and such approaches have been well studied (e.g., Planning with Diffusion for Flexible Behavior Synthesis). The advantage of injecting reward signals at noise steps, compared with such classifier-free guidance generation methods, should be clarified.

2.	The baselines in Table 1 include few recent flow or diffusion-based methods, and Diffuser is a 2022 work. More state-of-the-art diffusion-based methods from the past two years should be added for performance comparison.

3.	There is a lack of hyperparameter sensitivity analysis. Three hyperparameters are used in Table 3, but only the sensitivity of guidance length is analyzed in Figure 2.

4.	Figure 1 shows that training slows down as guidance strength increases. A deeper analysis should be provided here, including the change in computational complexity and its relationship with sample efficiency and performance.

5.	There is a significant gap in the derivation from Eq. 12 to Eq. 13. The paper should provide a more detailed derivation to bridge this transition.

6.	In Eq. 13, given that $\eta$ is already present, what is the specific role or necessity of introducing the additional parameter $\lambda$?

7.	Eq. 10 already incorporates a conservative constraint (KL divergence from the behavior policy $\pi_\beta$). Why is it necessary to apply additional conservatism to the value function using ensemble models in Eq. 14? Furthermore, since ensemble models often introduce substantial computational overhead, the experimental section should analyze the impact of this additional conservatism on both performance and computational cost.

8.	To avoid backpropagation through time for value gradients, FQL (Park et al., 2025) offers an alternative approach by distilling a multi-step model into a one-step model. This method should be compared and discussed in the paper.

9.	The fairness of the experimental comparison should be ensured. The AntMaze performance curves in Figure 4 are confusing; standard baseline practices typically involve evaluating over 100 episodes per round. Additionally, most baselines are typically trained for 1 million gradient steps, whereas this paper states that the proposed method is trained for 2 million gradient steps. Are the reported results for all methods compared under the same standard?

---

> ### Author Rebuttal · Authors · 2026-03-31
>
> We thank the reviewer for the thoughtful feedback. We appreciate the recognition of the theoretical foundation, and the effectiveness of our empirical results. We address the concerns below:
>
> ---
>
> ### (1) Motivation and relation to prior guidance methods
> We thank the reviewer for the opportunity to clarify our work (more details in the intro and related works.)
>
> First, prior methods like diffuser apply reward guidance at sampling time, while training remains behavior cloning. Deferring reward optimization to inference adds overhead and limits the formulation as an offline RL method.
>
> Second, action-space reweighting (e.g. AWR) can be data-inefficient and may underutilize the learned value function; as discussed in our paper (`sec. 3.3`) and supported by prior analyses [1], this limits policy learning.
>
> Third, methods such as Diffusion-QL require backpropagation through the denoising chain, increasing compute cost and introducing instability.
>
> In contrast, our approach addresses the mismatch between the noisy and the clean action space of value functions. By defining the RL objective directly in the noisy space, we enable an efficient policy gradient without backpropagating through the denoising process.
>
> ---
>
> ### (2 & 8) Baselines and comparison to recent methods (including FQL)
> FQL (2025) is discussed in the paper (page 1, last paragraph). We have now included it in our experimental comparison as reported below:
> |Agent|AntMaze-m-p|AntMaze-m-d|AntMaze-l-p|AntMaze-l-di|Pen-Human|Pen-Cloned|
> |-|-|-|-|-|-|-|
> |FQL|72|68|42|48|53|74|
> |NSAC(Ours)|92.1|89|77.8|78|83.9|84.6|
>
> Our method outperforms FQL across all tasks.
>
> ---
>
> ### (3, 4 & 7) Hyperparameters, compute cost, and guidance effects
>
> Guidance strength and training cost: Guidance strength does not affect training or inference time. The performance drop at high guidance (`Fig. 1`) reflects the expected distributional shift in offline RL when the policy deviates from the dataset.
>
> Ensemble & pessimism. While Eq. (10) enforces an (actor) policy constraint, pessimism (Eq. 14) addresses overestimation in the critic, a known issue in offline RL. The ensemble follows [2]; we ablate its size (`Fig. 6`) and will report computational cost in the revision.
>
> Monte Carlo estimation and cost, Hyperparameter sensitivity: We kindly refer the reviewer to comment (b) for reviewer bj8W.
>
> ---
>
> ### (5 & 6) Theoretical clarifications (Eq. 12→13 and role of λ)
> Derivation from Eq. (12) to Eq. (13): the derivation follows the standard score-matching formulation used in prior diffusion actor–critic methods [3], which we reference in the paper; we will make this clearer in the revision
>
> Role of λ in Eq. (13): λ controls a behavior cloning warm-up: it starts large and is gradually reduced during early training (similar to DTQL) to stabilize training.
>
> ---
>
> ### Evaluation protocol and fairness of training budget
> As detailed in the appendix, all results are reported after convergence without any online model selection. Specifically, we average checkpoints from the last 50k training steps, each evaluated over 10 rollouts, to obtain an accurate estimate. This follows the same evaluation used in prior works (e.g.: DAC)
>
> Regarding training budget, our setup is consistent with standard diffusion-based methods. For example, DAC is trained for 2M steps, Diffusion-QL for ~1–2M steps, and IDQL involves ~1.5M critic and up to 3M actor updates. Our training budget therefore lies well within the commonly adopted range. Importantly, as shown in `Fig. 4`, performance at 1M steps is comparable to 2M, indicating that our results do not rely on extended training.
>
> ---
> ### Limitations/Questions
>
> Clarification on relation to classifier-free guidance (CFG): CFG applies value guidance at sampling time, while our method integrates value optimization during training via a policy gradient in noisy space. Thus, CFG operates at inference time, whereas ours define a training objective
>
> Importantly, CFG introduces a semantic mismatch: the value function is defined over clean actions, while the policy operates in noisy space. Thus, guidance does not correspond to a well-defined policy gradient. Moreover, applying the Q-function through noisy latents can amplify extrapolation errors, worsening the OOD issue in RL. We consider this a core motivation for our work.
>
> In contrast, we define a noisy-space value function (via clean-action expectations) and derive its policy gradient, aligning value estimation with policy optimization
>
> This is a change in formulation—not a reparameterization—yielding a well-defined objective with improved efficiency, stability, and consistent gains (`Table 1`)
>
> ---
> [1] Is Value Learning Really the Main Bottleneck in Offline RL?
>
> [2] Why So Pessimistic? Estimating Uncertainties for Offline RL through Ensembles, and Why Their Independence Matters
>
> [3] Diffusion Actor-Critic: Formulating Constrained Policy Iteration as Diffusion Noise Regression for Offline Reinforcement Learning

---

> > ### Author Rebuttal · Reviewer_oMgq · 2026-04-02
> >
> > Thank you for the clarifications. However, some concerns regarding related methods remain:
> >
> > 1. Estimate noisy space value functions: BDPO [1] also estimates value functions within intermediate diffusion steps. To properly situate your contribution, the paper should discuss its conceptual differences from BDPO.
> >
> > 2. Entirely avoids noisy-space guidance: The paper briefly mentions FQL but lacks a discussion on the strengths and weaknesses of both methods. Furthermore, your reported FQL scores are quite different from those reported in other papers (FQL and [2]), and this gap remains unexplained
> >
> > [1] Gao, Chen-Xiao, et al. "Behavior-Regularized Diffusion Policy Optimization for Offline Reinforcement Learning."
> >
> > [2] Zhang, Songyuan, et al. "ReFORM: Reflected Flows for On-support Offline RL via Noise Manipulation."

---

> > > ### Author Response · Authors · 2026-04-02
> > >
> > > We thank the reviewer for the thoughtful comments and for raising these points. We address them below:
> > >
> > > ---
> > >
> > > > BDPO [1] also estimates value functions within intermediate diffusion steps. To properly situate your contribution, the paper should discuss its conceptual differences from BDPO.
> > >
> > > We thank the reviewer for bringing up this relevant connection. BDPO [1] introduces value functions at intermediate diffusion steps (Eq. 15). However, their role is fundamentally different from our formulation. In BDPO, these value functions estimate cumulative diffusion penalties arising from KL regularization along the trajectory, rather than expected environment returns.
> > >
> > > In contrast, our noisy-space Q-function assigns to each latent variable the expected return of the actions it induces, yielding a value function that is directly aligned with the RL MDP objective. Importantly, BDPO’s intermediate value functions serve as auxiliary constructs for optimizing diffusion dynamics, whereas our formulation defines a value function that is consistent with the underlying MDP semantics. This distinction is crucial, as it enables well-defined policy gradients in noisy space without evaluating critics on noisy, out-of-distribution variables. This leads to a fundamentally different objective and optimization procedure. We will expand this discussion in the revised version of the paper.
> > >
> > > ---
> > >
> > > > The paper briefly mentions FQL but lacks a discussion on the strengths and weaknesses of both methods.
> > >
> > > We thank the reviewer for highlighting FQL [2]. FQL avoids applying value guidance in the noisy latent space by instead training a separate one-step policy that maximizes Q-values, while regularizing it via distillation from a generative model.
> > >
> > > In contrast, our approach directly addresses the core challenge of value-based optimization for diffusion policies by defining a value function over noisy latents that is aligned with action-space returns. This enables principled policy gradients at each denoising step without requiring distillation or restricting optimization to a one-step policy.
> > >
> > > Thus, FQL avoids optimizing the diffusion policy directly and instead learns an additional one-step policy, whereas our method provides a principled formulation for performing value-based optimization within the diffusion policy itself. We will expand this discussion in the revised version.
> > >
> > > ---
> > >
> > > > Furthermore, your reported FQL scores are quite different from those reported in other papers (FQL and [2]), and this gap remains unexplained.
> > >
> > > We thank the reviewer for raising this point. For the **Adroit tasks (Pen-Human-v1 and Pen-Cloned-v1)**, we report the same results as those presented in the FQL paper (i.e., taken directly from their reported results), and therefore the values are directly comparable. This also applies to the visual manipulation tasks from the OGBench benchmark requested by Reviewer 5A1x.
> > >
> > > For AntMaze, it is important to note that **some works (e.g., FQL [2], ReFORM [3]) report results on different dataset versions (e.g., v0 vs v2)**, which are not directly comparable and often lead to significant differences in reported performance. This **dataset version mismatch has also been noted in prior work (e.g., DTQL [4], Remark 4, Page 8)**. In our experiments, we consistently use the v0 benchmarks, which are also adopted by prior methods (e.g., DAC, DTQL, IQL, DQL), ensuring fair comparison.
> > >
> > > If the reviewer has a specific environment or result in mind where a discrepancy appears, we would greatly appreciate clarification so we can address it directly.
> > >
> > > ---
> > >
> > > ###### [1] Gao, Chen-Xiao, et al. *Behavior-Regularized Diffusion Policy Optimization for Offline Reinforcement Learning.*
> > >
> > > ###### [2] Park, Seohong, et al. *Flow Q-Learning.*
> > >
> > > ###### [3] Zhang, Songyuan, et al. *ReFORM: Reflected Flows for On-support Offline RL via Noise Manipulation.*
> > >
> > > ###### [4] Chen, Tianyu, et al. *Diffusion policies creating a trust region for offline reinforcement learning.* Advances in Neural Information Processing Systems 37 (2024).

---

### Official Review · Reviewer_UvqV · 2026-03-12

**Soundness:** 4
**Presentation:** 3
**Significance:** 4
**Originality:** 4
**Overall Recommendation:** 5
**Confidence:** 4

**Summary:**

The paper proposes Noisy-Space Policy Gradient (NSPG), a framework for training diffusion policies with value-based reinforcement learning. The authors focus on the "space mismatch" between diffusion training (latent-noise space) and the MDP-interface (e.g., actions, values). To address this, NSPG models a state-action value directly in the diffusion noise space, which ultimately allows one to perform policy improvement without needing to backprop through the diffusion denoising process.

The authors benchmark NSPG and a handfull of other offline RL algorithms in several d4rl locomotion and antmaze tasks, and two of the manipulation tasks from adroit. The results show that NSPG is always at the top in terms of performance. The authors also ablate over two impactful design choices: the guidance strength and the number of Monte Carlo samples.

**Compliance With Llm Reviewing Policy:**

Affirmed.

**Key Questions For Authors:**

I  am recommending acceptance, so my questions/requests are minor.

1. The authors should include a diagram that displays the intuition behind their proposed approach.
2. The authors should cite and compare to the relevant paper ReinFlow (Zhang et al., 2025).
3. Should the x-axes of Fig 2 & 3 be "Gradient Updates" instead of "Environment Steps"?

**Limitations:**

yes

**Strengths And Weaknesses:**

### Soundness
**Strengths**
- The claims made in the paper  are supported by both empirical and theoretical evidence.
- The experiments are standard for the offline RL setting.
- The practical implementation of the theoretical framework is reasonable.

**Weaknesses**
- None

### Presentation
**Strengths**
- The authors do a nice job of walking the reader through the idea and providing proofs alongside intuition.
- I am a big fan of the "Interpretation" box on page 4.
- I appreciate the explicit comparison to AWR in $\S3.3$.

**Weaknesses**
- A diagram presenting the method and the intuition behind how it works would be a nice addition to the paper.



### Significance
**Strengths**
- Guiding powerful, generative models is useful in many scenarios. The authors cover one of particular interest (generating a controller), but the usage of advantage could be applied to a wide array of downstream applications.

**Weaknesses**
- None

### Originality
**Strengths**
- To the best of my knowledge, the noisy-space view of the value function is novel. Defining a value-learning signal over intermediate diffusion states, rather than only over final actions, is distinct from the usual way these methods are presented.
- The authors do a fine job in differentiating their approach from other, relevant, methods.

**Weaknesses**
- None

---

> ### Author Rebuttal · Authors · 2026-03-31
>
> ## Rebuttal to Reviewer UvqV
>
> We thank the reviewer for the detailed and constructive feedback. We appreciate the positive assessment of the paper’s technical soundness, empirical validation, and the novelty of the noisy-space formulation. We are also encouraged by the positive feedback on the presentation, including the clarity of the exposition, the usefulness of the “Interpretation” box, and the comparison to AWR in Section 3.3.
>
> Below we address the suggestions:
>
> ---
>
> ### 1. Missing diagram for intuition
>
> We agree that a visual schematic would improve clarity. In the revision, we will include a diagram in the main paper illustrating:
> - the relationship between noisy latent space and action space,
> - how the noisy-space value is defined,
> - and how policy improvement operates without backpropagating through denoising.
>
> We also note that the appendix already includes a visualization of the noisy-space gradient field (`Fig. 3`), comparing NSPG with a naive decoded gradient. This figure highlights the core intuition behind our method. We acknowledge that it is currently under-emphasized, and will:
> - explicitly reference it in the main text,
> - improve its explanation,
> - and better connect it to the method.
>
> ### 2. Comparison to ReinFlow (Zhang et al., 2025)
>
> We thank the reviewer very much for highlighting this relevant work. ReinFlow is a strong contribution that enables stable **online RL for flow policies** by introducing noise to obtain a tractable Markov formulation and exact likelihoods for policy gradients.
>
> Our work addresses a complementary setting: **offline RL for diffusion policies**, focusing on the mismatch between noisy latent optimization and action-space value. We resolve this by defining a **noisy-space value function grounded in action returns** and deriving a corresponding policy gradient that avoids both critic evaluation on noisy latents and backpropagation through denoising.
>
> We will add a discussion to ReinFlow to clarify these distinctions and better position our contribution.
>
> ### 3. Clarification of x-axes in Figures 2 & 3
>
> We thank the reviewer for spotting this. Since our method operates in the offline RL setting, the x-axes should reflect **training/gradient steps**, not environment steps. We thank the reviewer very much for the attentive reading, and will correct the labels and ensure consistency in the figure.
>
> ---
>
> We thank the reviewer again for the helpful suggestions and the thoughtful feedback. We believe these changes will improve clarity and positioning. We remain fully open to address any comments or questions the reviewer may have.

---

### Decision · Program_Chairs · 2026-04-30

**Decision:**

Accept (regular)

**Comment:**

This paper tackles the conceptual and algorithmic challenges of integrating diffusion policies into offline RL. The primary innovation is the introduction of a noisy-space action-value function. This function assigns values to diffusion latents by considering the distribution of intermediate actions induced by the denoising process. This construction provides a precise semantic interpretation. Building on this, the authors derive a Noisy-Space Policy Gradient (NSPG), which computes value estimates for noisy latents exclusively using clean action-space values. Furthermore, they formulate a KL-regularized policy improvement over noisy latents, showing that the resulting objective admits a diffusion-compatible regression form, successfully avoiding the need for backpropagation through the complex denoising process.

The paper presents a principled and effective foundation for training diffusion policies in the offline RL setting. By developing the NSPG, the authors successfully navigate the algorithmic complexities associated with using diffusion models for continuous control under offline constraints. The experimental results on the D4RL benchmark demonstrate that these semantically grounded value gradients provide a robust and effective method for policy training. Given the novelty of the noisy-space construction and its successful application to a challenging domain, I recommend acceptance.